# Human ACE2 transgenic pigs are susceptible to SARS-CoV-2 and develop COVID-19-like disease

Long Fung Chau [1,6], Simon Lillico[1,6], Tanja Opriessnig[2], Rosemary Blake[1], Luc Tardy[1], Chen-Hsuin Lee [1], Scott Maxwell[1], Claire Warren[1], Elizabeth Thornton[1], Catherine L. Mclaughlin[1], Gerry McLachlan[1], Christine Tait-Burkard [1], Sarah Fletcher [1], Stephen Anderson[2], Sharon Brown[2], Louise Gibbard [2], Thomas Tzelos[2], Dawn MacMillan-Christensen[2], J. Kenneth Baillie [1,3], David A. Dorward[4,5], David J. Griffiths [2] ✉ & Finn Grey [1] ✉

Animal models that accurately reflect COVID-19 are vital for understanding mechanisms of disease and advancing development of improved vaccines and therapeutics. Pigs are increasingly recognized as valuable models for human disease due to their genetic, anatomical, physiological, and immunological similarities to humans, and they present a more ethically viable alternative to non-human primates. However, pigs are not susceptible to SARS-CoV-2 infection which limits their utility as a model. To address this, we have developed transgenic pigs expressing human ACE2 that are susceptible to SARS-CoV-2 infection. Following challenge, clinical signs consistent with COVID-19, including fever, coughing and respiratory distress were observed, with virus replication detected in the nasal turbinates, trachea and lungs up to the study endpoint, seven days post-infection. Notably, examination of tissues revealed immunopathology in the lungs consistent with histological changes observed in fatal human COVID-19 cases. This study establishes human ACE2 transgenic pigs as a large animal model that accurately reflects many aspects of COVID-19 disease.

The key molecular events that trigger progression from self-limited viral illness to severe COVID-19 remain poorly defined. Non-invasive profiling of cells from patients and analysis of tissue specimens collected *post-mortem* has been valuable in characterising inflammatory pathology associated with severe COVID-19[1,2]. However, defining the key early events that trigger progression towards severe COVID-19 requires longitudinal studies and invasive sampling of tissues.

Animal models are important tools for testing hypotheses about mechanisms of disease in COVID-19 and complement clinical studies from patients, enabling direct analysis of target tissues following controlled infections[3]. However, at present, there is a lack of efficient, tractable model systems that replicate the primary features of severe COVID-19. Human disease associated with COVID-19 can be broadly categorised as mild, moderate and severe based on clinical symptoms[4–6]. Mild disease is associated with fever, fatigue, cough and

[1]Roslin Institute and Royal (Dick) School of Veterinary Studies, University of Edinburgh, Edinburgh, UK. [2]Moredun Research Institute, Edinburgh, UK. [3]Baillie Gifford Pandemic Science Hub, University of Edinburgh, Edinburgh, UK. [4]Centre for Inflammation Research, Queen's Medical Research Institute, Edinburgh, UK. [5]Department of Pathology, Royal Infirmary, Edinburgh, UK. [6]These authors contributed equally: Long Fung Chau, Simon Lillico. ✉e-mail: David.Griffiths@moredun.ac.uk; fgrey@exseed.ed.ac.uk

sore throat, with minimal or no visible lesions in the lungs based on chest radiographs. Moderate disease includes respiratory distress, with blood oxygen saturation levels between 90 and 94% and signs of mild lung infiltrates. Severe disease is associated with dyspnoea, high respiratory rate and oxygen saturation levels below 90%. The histological changes observed within the lungs of patients who have died from COVID-19 can be variable but include exudative and organising phase diffuse alveolar damage (DAD), which includes hyaline membranes, oedema, inflammatory cell infiltrates, intra-alveolar fibrin, organising pneumonia, suppurative bronchopneumonia and intravascular thrombi[1].

Rodent models such as human angiotensin-converting enzyme 2 (hACE2) expressing transgenic mice and golden Syrian hamsters are susceptible to SARS-CoV-2 infection, while non-transgenic mice can be infected with mouse-adapted strains of SARS-CoV-2. Ferrets are also susceptible to infection and are effective transmission models, while non-human primates (NHPs) are the most comparable model to human infections[3]. While each of these models has advantages, there are important drawbacks. Rodent models do not accurately replicate the pattern of disease, tissue morphology and immunological responses to SARS-CoV-2 in humans[3]. There is a lack of molecular tools for ferret models, while non-human primates are expensive and restricted to a few institutions around the world. The lack of large animal models that accurately reflect the pathology associated with severe COVID-19 in humans hampers our ability to understand the mechanisms that drive disease and the development of effective interventions. Furthermore, the development of therapeutic interventions in a model with similar physiology to humans is more likely to successfully translate into effective therapies in humans.

There is an increasing appreciation of livestock as biomedical models, with pigs being one of the most important[7]. The short gestation period, large litter size and extensive genome editing tools, allow rapid development of transgenic large animal models[8]. Pigs are considered the species of choice for xenotransplantation, reflecting their similarity to human anatomy and physiology[9]. The upper airway, lungs, heart and brain, all targets of SARS-CoV-2 infection, are physiologically and anatomically more similar to humans than those of rodents and ferrets[9,10]. Critically, for vaccine development and studies on immune pathology, the porcine immune system more closely resembles humans in greater than 80% of parameters analysed, compared to less than 10% for mice[11]. This includes a higher percentage of neutrophils in the blood of pigs and humans than in mice[12]; expression of CXCL8, a chemoattractant for neutrophils, present in pigs and humans, but absent in mice[12]; and key differences in dendritic cells in mice compared to humans and pigs which leads to altered responses to innate antagonists often used as vaccine adjuvants[13].

Vaccines can be administered intramuscularly, subcutaneously, intradermally, orally or intranasally and pigs can be routinely bled and immunized using well-established protocols. Compared to rodents, large numbers of immune cells can be isolated and pigs offer easy access to various immune compartments[7]. Various surgical and non-surgical procedures typically used in human medicine can be performed in pigs, including catheterization, heart surgery, valve manipulation, endoscopy and bronchoalveolar lavages. These procedures are particularly difficult or impossible to perform in smaller animal models including rodents. Pigs are relatively inexpensive, accessible and are more ethically acceptable than NHPs.

However, multiple studies have shown that pigs are not susceptible to SARS-CoV-2 infection which limits their utility as a model[14,15]. To address this, we generated transgenic pigs expressing hACE2, the primary cellular receptor for SARS-CoV-2 and the main determinant of species tropism[16,17]. We showed that the pigs were susceptible to infection with SARS-CoV-2 through intranasal inoculation and displayed clinical signs consistent with COVID-19 in humans, including fever, coughing and respiratory distress. Crucially, histological analysis demonstrated significant inflammation in the lungs, consistent with pathology seen in fatal COVID-19 patients. These findings demonstrate that hACE2 transgenic pigs represent a large animal model that accurately reflects pathological disease of COVID-19 in humans and represents a valuable model for studying disease pathology, vaccines and novel therapeutics.

## Results

### Generation of hACE2 expressing transgenic pigs

To generate the transgenic pigs, a custom lentivirus expressing hACE2 under the Keratin 18 promoter was microinjected into the perivitelline space of putative zygotes, which were then surgically implanted into five surrogate gilts (Fig. 1). Three gilts were confirmed pregnant and a total of 32 piglets were born. Genomic DNA was generated from ear biopsies and relative lentivirus copy number determined using a provirus-specific qPCR, with cycle threshold (Ct) values ranging from 19.7 to 30.7 (Supplemental Table 1). Based on Ct values, three females and two males were selected for breeding, generating an F1 cohort of 30 piglets. One piglet died three days after birth (P41), leaving a cohort of 29 piglets. Total RNA was extracted from ear biopsies and levels of hACE2 transcript was determined by RT-qPCR. Piglets were ranked based on hACE2 transcript levels, with the nine highest expressing piglets (seven females and two males) selected for the challenge study with SARS-CoV-2 (Supplemental Table 2).

### hACE2 primary fibroblasts are susceptible to SARS-CoV-2 infection in vitro

Prior to the challenge study, primary fibroblast cells were generated from ear biopsies taken from all nine selected pigs and an additional two transgenic pigs that showed low or undetectable levels of transgene expression (P35 and P38). We were unable to recover cells from

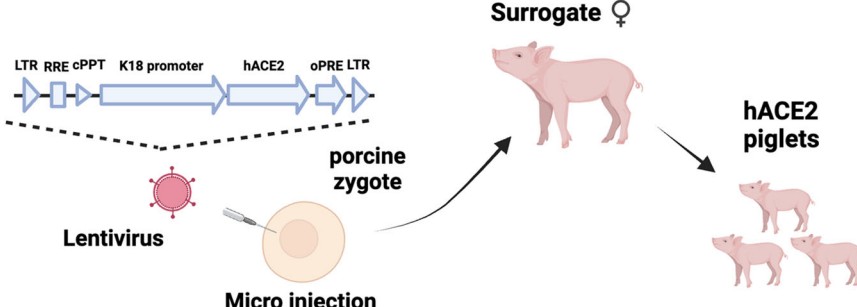

**Fig. 1 | Schematic representation of generation of hACE2 expressing transgenic pigs.** A custom lentiviral vector with the human K18 promoter driving hACE2 expression was microinjected into the perivitelline space of putative zygotes. Zygotes were then implanted into surrogate gilts (Created in BioRender. Grey, F. (2023) https://BioRender.com/l58r269).

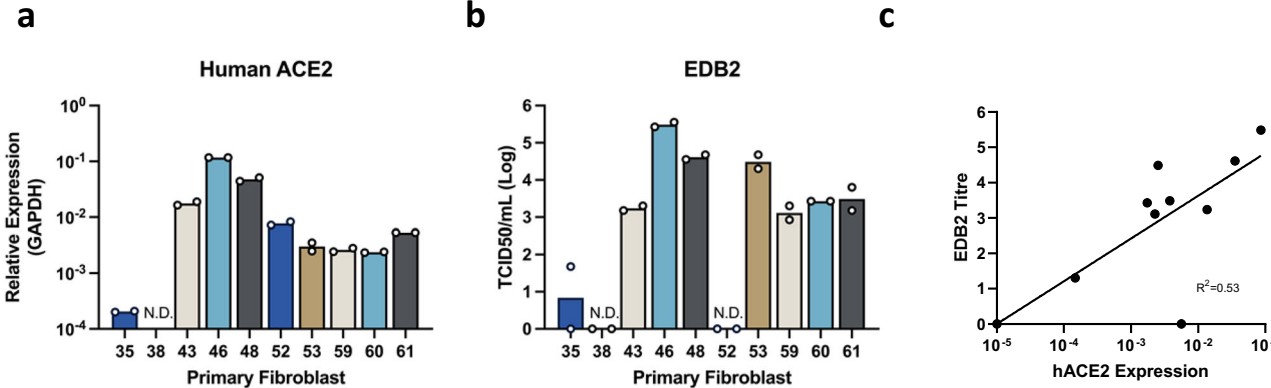

**Fig. 2 | hACE2 transcript levels correlate with susceptibility to SARS-CoV-2.**
**a** Primary cells were generated from ear notches taken from transgenic pigs. Total RNA was harvested and hACE2 transcript levels determined by RT-qPCR. hACE2 levels were normalized to GAPDH and shown as relative expression. **b** To test susceptibility, fibroblast cells were infected at an MOI of 3. Supernatant was harvested 24 HPI and infectious virus quantified by $TCID_{50}$. Data represents the mean from two independent experiments (each using two technical replicates). ND = not detected. **c** Correlation of SARS-CoV-2 titre and hACE2 expression levels are shown. Source data are provided as a Source Data file.

one of the biopsy samples (P57) due to bacterial contamination. RT-qPCR analysis revealed that hACE2 mRNA levels in individual cell lines were similar to the associated ear biopsies (Fig. 2a). To determine susceptibility, cells were infected with an early pandemic isolate of SARS-CoV-2 (EDB2)[18] and viral titres determined at 24 h post infection (HPI) by Tissue Culture Infectious Dose 50 ($TCID_{50}$) assays. The results showed that, in all but one case, cell lines expressing higher levels of hACE2 mRNA were more susceptible to SARS-CoV-2 infection in vitro, based on higher titres of virus being produced (Fig. 2b, c).

A low level of virus replication was detected in cells from P35 and no virus was detected from cells from P38, consistent with the low levels of hACE2 detected in the tissues and cells from these pigs. Surprisingly, no virus replication was observed in cell line P52, despite high levels of hACE2 expression being detected. It is currently unclear why this cell line was unable to support SARS-CoV-2 virus replication. The cells were also infected with SARS-CoV-2 Delta and Omicron to determine susceptibility to more recently emerged viral variants (Supplemental Fig. 1). As Omicron does not induce clear cytotoxic effects in cell lines, viral genome levels in the supernatant were measured using RT-qPCR. Similar to EDB2, replication of Delta virus correlated with hACE2 expression levels. In contrast, Omicron replication was only observed in the two cell lines derived from P46 and P48 which have the highest hACE2 expression levels, indicating susceptibility to more recent strains of SARS-CoV-2 is dependent on higher levels of hACE2 expression. We further confirmed the susceptibility of selected cell lines by immunofluorescent staining for SARS-CoV-2 N protein, showing all three variants able to infect P46 and P48 cell lines, while none were able to infect P52 (Supplemental Fig. 1d). Omicron was able to infect P46 and P48, but no N protein was detected in the other cell lines (P43 shown as example), despite expressing hACE2 and being susceptible to EDB2 and Delta. Further studies will be required to understand why the cells generated from P52 were not susceptible to SARS-CoV-2 infection.

## hACE2 pigs are susceptible to SARS-CoV-2 infection and display COVID-19-like clinical signs

To determine in vivo susceptibility, the nine selected transgenic pigs, and three genetically similar non-transgenic controls (referred to from here on as WT) were challenged at biosafety level three on a single occasion with $1 \times 10^6$ $TCID_{50}$ of EDB2. EDB2 was selected for the challenge studies as it efficiently infected primary fibroblast cells from the transgenic pigs and early isolates were associated with more severe disease, therefore more likely to generate clear clinical and histological outcomes in the challenge study. The inoculum was

delivered intranasally in a single 2 ml dose using a mucosal atomiser attached to a syringe. The dose and route of infection were based on previous challenge studies in pigs using SARS-CoV-2 or the porcine adapted coronavirus PRCV[19]. Rectal temperature and clinical status of the pigs were monitored twice daily and SARS-CoV-2 lateral flow tests (LFT) performed on nasal swabs collected two, four and seven days post infection (DPI). Three transgenic pigs and one WT pig were euthanised at two, four and seven DPI (cohort 1, 2 and 3, respectively) with tissues, including the nasal turbinates, tracheal epithelium (proximal, mid and distal) and lung, collected for virus detection and histological analysis.

Within 24 h of infection P43, P46, P48, P60 and P61 all showed signs of fever, with temperatures above 40 °C, although P43 and P48 showed signs of elevated temperature before infection, suggesting the initial fever may not be directly related to SARS-CoV-2 infection (Fig. 3a–c). During the course of the experiment, only P57 and the WT control pigs showed no signs of fever at any time point. However, as part of cohort one, P57 may have developed fever at a later stage of infection had it not been culled two DPI. All transgenic pigs tested positive by LFT of nasal swabs, as early as two DPI (Fig. 3d–f), confirming their susceptibility to SARS-CoV-2 infection. In contrast, control WT pigs were negative by LFT throughout the challenge study.

In addition to fever and positive LFT tests, the transgenic pigs displayed other clinical signs consistent with COVID-19, including sneezing, coughing and respiratory distress. A clinical assessment scheme was developed to score each animal against several criteria, including demeanour (e.g. response to stimulation, response to presence of food, willingness/ability to stand when provoked), appetite (interest/enthusiasm in feeding/drinking), respiration (degree of effort in respiration, presence/absence of nasal discharge), and other respiratory signs (presence and duration and continuity of coughing and/or sneezing). A range in clinical severity was observed over the course of the infection and between individual pigs and are summarised in Supplemental Table 3. At early time points (by 48 HPI) occasional coughing and mild respiratory distress were observed in some hACE2 pigs (P46, P48, P53 and P61). However, by 96 HPI moderate clinical signs were observed. In particular, P46 showed signs of lethargy, reluctance to stand, extended intermittent periods of coughing, laboured respiration and nasal discharge. Cumulative clinical assessment scores were calculated to enable direct comparison of animals over the time course (Fig. 3g), indicating that P46 and P60 displayed the most severe clinical signs before being culled 4 DPI.

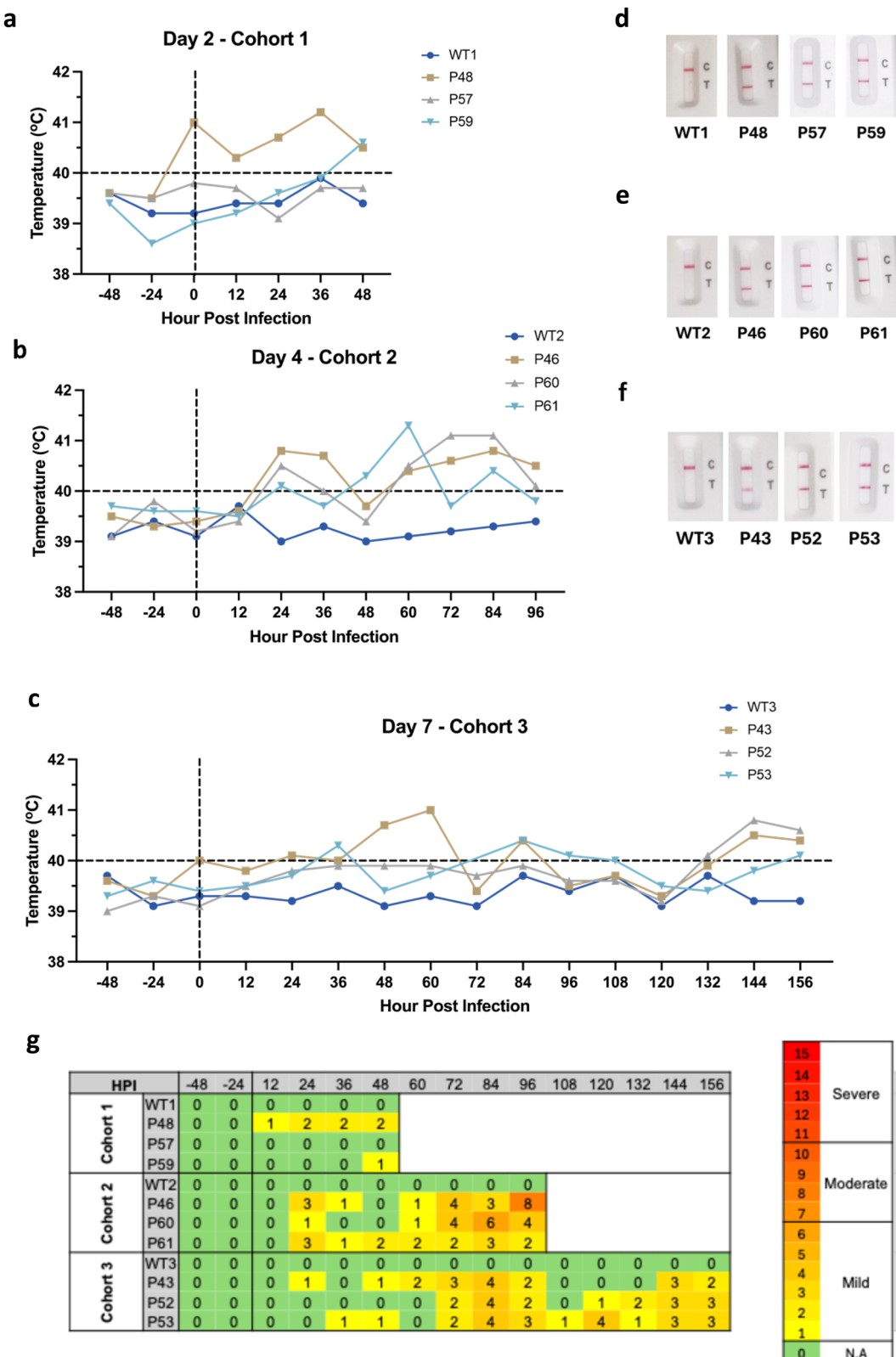

**Fig. 3 | Infection with SARS-CoV-2 results in clinical signs consistent with COVID-19. a** Rectal temperatures of pigs infected with SARS-CoV-2. Data grouped according to cohorts culled at 2 (**a**), 4 (**b**) and 7 (**c**) DPI. **d**–**f** Results of SARS-CoV-2 LFTs grouped by DPI. C indicates control line; T indicates test line and indicates a positive for SARS-CoV-2 infection. All hACE2 pigs were positive, while all WT pigs were negative. **g** Summary of clinical scores for human ACE2 infected pigs. A clinical assessment scheme was established prior to the study, in which pigs were scored against several criteria including demeanor, appetite, temperature, respiration and other respiratory signs (cough, sneeze). Pigs were scored daily and the cumulative score recorded. Increases in temperature and respiratory stress were recorded in all hACE2-transgenic pigs except P57 but not in the wild-type animals. Cumulative scores of 7 or greater are defined as moderate severity in this model. HPI hours post infection. Source data are provided as a Source Data file.

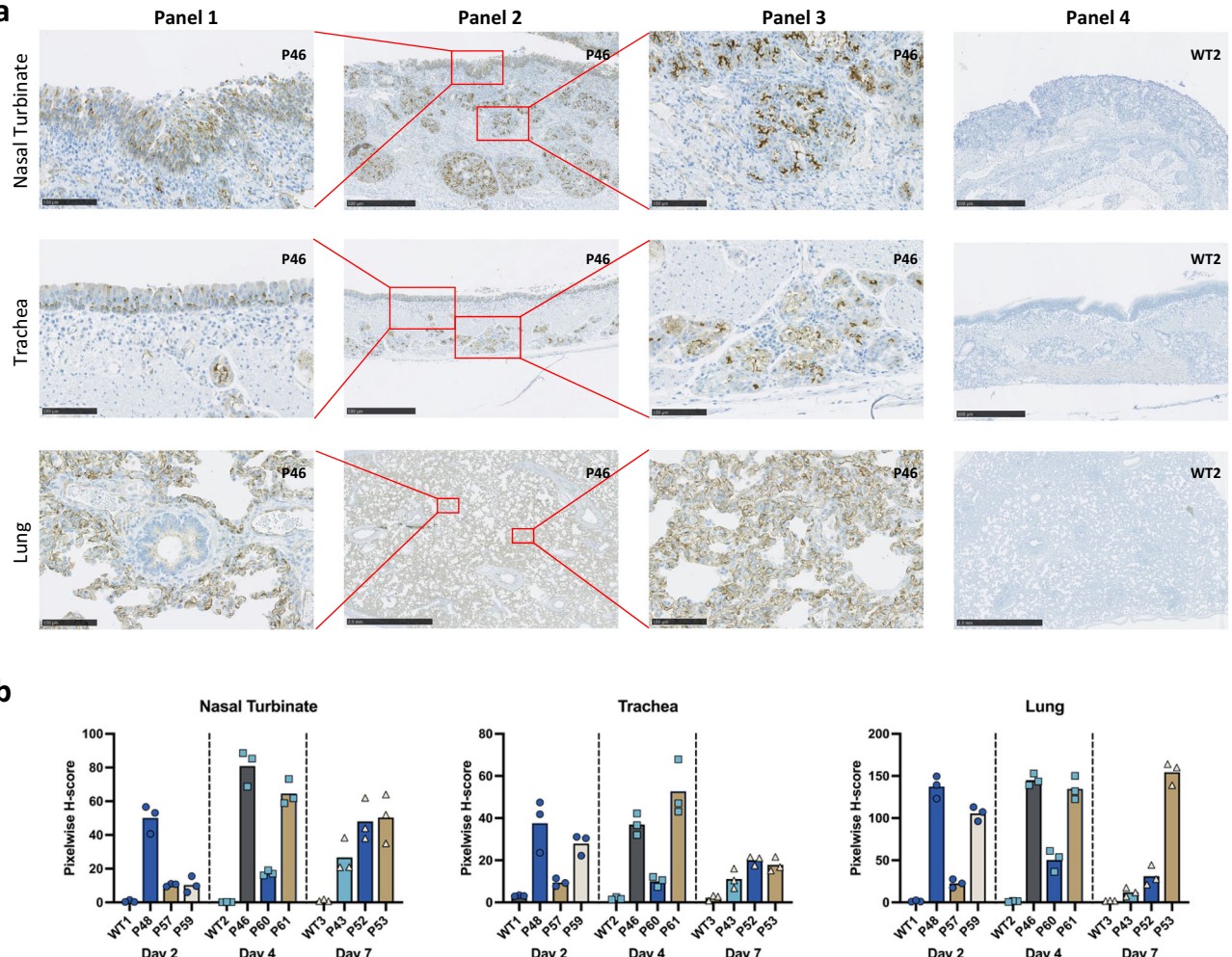

**Fig. 4 | Expression and distribution of hACE2 in pig tissues. a** IHC staining demonstrating distribution of hACE2 expression in sections of nasal turbinate, trachea, and lung from P46 culled four DPI. Red boxes in panel 2 indicate the location of magnified images displayed in panels 1 and 3. Panel 4 shows the control WT pigs without hACE2 expression. Staining indicates hACE2 expression in ciliated columnar epithelial cells, goblet cells and seromucinous glands within the nasal turbinates and trachea. In lungs, staining is most pronounced in alveolar pneumocytes but to a lesser extent in bronchiolar epithelium and vascular endothelium.

(Panel 1, 3 and 4 −5x magnification, (except lung − 1x); panel 2 and 4 − 20x magnification). Scale bars for panel 1 and 3 are 100 µm; for panel 2 and 4 are 500 µm, apart from lung images which are 2.5 mm. **b** Total hACE2 staining was quantified using QuPath software. Viable tissue regions on each whole-slide IHC image were randomly sampled three times by non-overlapping regions with areas no smaller than 4 mm². The pixelwise H-score for each region was calculated (see Methods) and the average score presented for each image. Source data are provided as a Source Data file.

## Distribution of hACE2 expression is similar between animals while expression levels vary

Expression levels and tissue distribution of ACE2 plays an important role in determining susceptibility to SARS-CoV-2 infection. In humans, single-cell RNAseq analysis and immunohistochemistry (IHC) have identified widespread expression of ACE2 in multiple organs, tissue and cell types[20–22]. In the upper respiratory tract, hACE2 is expressed in ciliated columnar epithelial cells and goblet cells. In the lungs, a small percentage of type II pneumocytes are thought to be the major target of SARS-CoV-2, where both ACE2 and the entry factor TMPRSS2 are co-expressed.

IHC was employed to determine hACE2 expression levels in the transgenic pigs. Nasal turbinate, trachea and lung samples, representing the upper and lower respiratory tract, were taken from transgenic and WT animals at two, four and seven DPI. Figure 4a shows images of hACE2 staining in tissues from P46 (a pig with high hACE2 expression) with representative images of tissues from all animals shown in Supplemental Fig. 2.

The observed expression pattern across tissues aligns with the expected epithelial distribution, consistent with the use of the K18 promoter to drive hACE2 transgene expression. Expression of hACE2 in the nasal turbinates and trachea appears to be confined to ciliated columnar epithelial cells, goblet cells and seromucinous glands. Expression in the lung parenchyma is widespread in alveolar pneumocytes, bronchiolar epithelium and vascular endothelium. However, further validation through co-staining or single-cell RNA sequencing will be necessary to definitively characterize the specific cellular profiles of hACE2 expression in the transgenic pigs.

While the pattern of hACE2 expression in transgenic pigs generally reflects that of ACE2 in human tissues, it appears more widely distributed in the pigs. Subjectively, a higher proportion of epithelial cells express hACE2 compared to similar human tissue expression[20–22]. Expression levels of hACE2 also vary considerably between the individual transgenic pigs, as would be expected when using a lentiviral transgenic approach, due to differences in copy number and integration site.

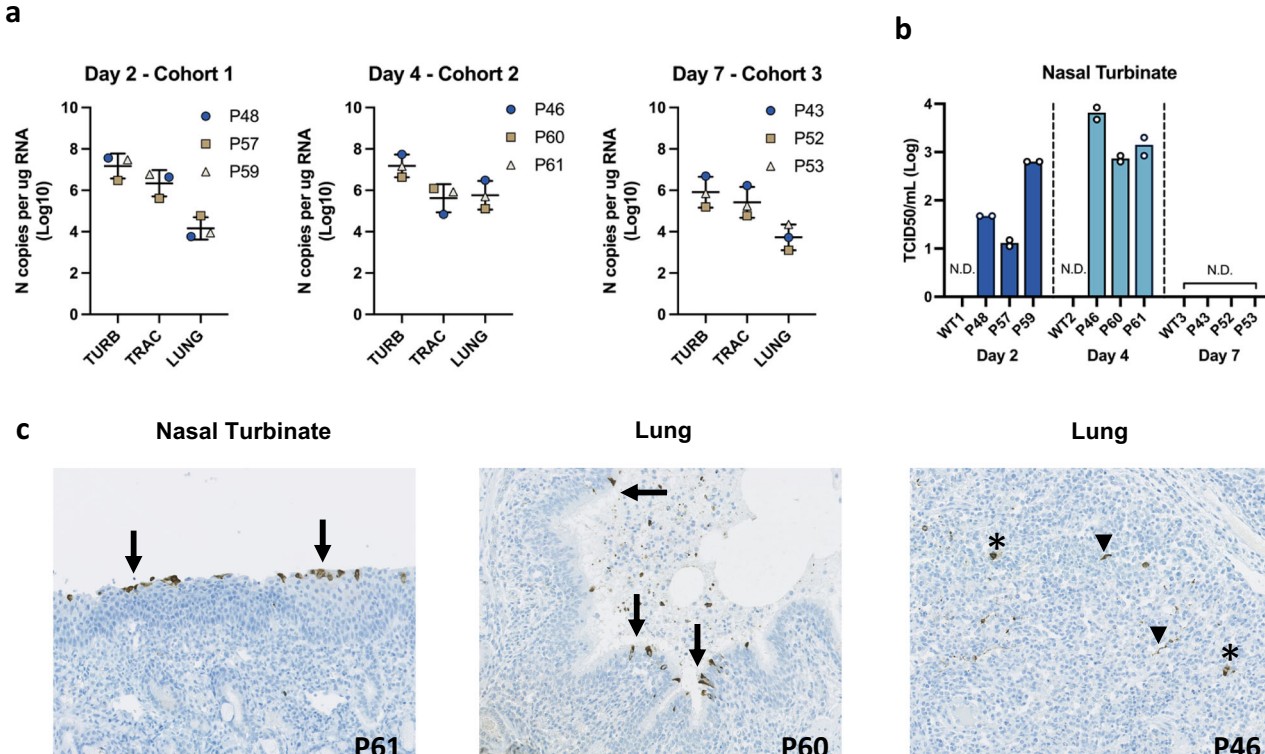

**Fig. 5 | SARS-CoV-2 virus load in respiratory tissue. a** SARS-CoV-2 viral RNA levels determined by RT-qPCR at two, four and seven DPI from TURB = nasal turbinates, TRAC = trachea, LUNG = lung. No viral RNA was detected in control WT pigs. P43–P61 indicates individual pig identifier corresponding to each data point (each with two technical replicates). Data represent the mean ± Standard Deviation (S.D.). **b** Levels of infectious virus from nasal turbinates were determined by TCID$_{50}$. No infectious virus was detected in trachea or lung. Data represents the mean from two independent experiments (each using two technical replicates). Source data are provided as a Source Data file. **c** Tissue sections from nasal turbinates and lung were stained for SARS-CoV-2 nucleocapsid protein demonstrating the presence of viral protein within mucosal epithelial cells of both the upper and lower respiratory tract (arrow) as well as alveolar macrophages (*) and pneumocytes (arrowhead). Representative images shown from hACE2 transgenic pigs. (20x magnification).

QuPath image analysis software was used to quantify hACE2 expression in the sections shown in Supplemental Fig. 2 (Fig. 4b). Expression levels were largely proportional between tissues of individual pigs, apart from P53 which displayed relatively low levels of expression in the trachea compared to the nasal turbinates and lung, and P59 which showed relatively low levels of hACE2 in the nasal turbinates compared to the trachea and lung. No staining was detected in any tissues from WT animals. Low ACE2 expression in the respiratory epithelium has been suggested as a reason why pigs are not naturally susceptible to SARS-CoV-2[23]. Elevated hACE2 expression in the respiratory epithelium of the transgenic pigs likely accounts for the increased susceptibility to the virus. However, other host determinants likely play a role in the severity of disease in individual animals.

### Robust replication of SARS-CoV-2 in hACE2 pigs

In humans, SARS-CoV-2 can replicate to high levels in the upper respiratory tract, enabling efficient spread through airborne transmission. In contrast, viral load is often lower in the lung and does not necessarily correlate with severe disease, as pathology in the lung is largely driven by dysregulated host immune responses rather than viral replication itself[1].

To determine the viral load in the upper and lower respiratory tract of hACE2 pigs, total RNA was isolated from nasal turbinate, trachea and lung tissue homogenates taken from transgenic and WT control animals. RT-qPCR analysis using a primer probe set targeting the viral N gene revealed high levels of viral RNA, especially in the nasal turbinates, at two and four DPI, ranging from $3.0 \times 10^6$ to $5.0 \times 10^7$ copies/μg total RNA (Fig. 5a). Levels in the trachea and lung

were lower, ranging between $5.9 \times 10^4$ and $5.9 \times 10^6$ copies/μg total RNA for trachea and $1.3 \times 10^3$ and $3.0 \times 10^6$ copies/μg total RNA for the lung.

Infectious virus was measured using TCID$_{50}$ assays to confirm active infection and virus replication was occurring in the transgenic pigs. Figure 5b shows infectious virus was recovered from the nasal turbinates with titres peaking at four DPI but was undetectable by seven DPI. No infectious virus was recovered from the trachea or lung samples from any animals. No viral RNA or infectious virus was detected in tissues taken from the WT control pigs, consistent with previous reports that non-transgenic pigs are not susceptible to SARS-CoV-2[14,15]. IHC staining identified SARS-CoV-2 nucleocapsid protein within the cytoplasm of respiratory epithelia of nasal turbinates (Fig. 5c). Within the distal lung parenchyma, nucleocapsid protein was identified in bronchiolar epithelial cells, macrophages, alveolar pneumocytes and occasional vascular endothelial cells (Fig. 5c). No SARS-CoV-2 nucleocapsid was detected in WT pig tissues.

### Inflammatory immunopathology in lungs of hACE2 pigs is consistent with severe COVID-19

Lung pathology in fatal COVID-19 is characterised by the spectrum of exudative and organising DAD together with pulmonary oedema, intra-alveolar fibrin and intravascular thrombosis[3]. This occurs along with extensive immune cell infiltrate in alveolar spaces and the interstitium by neutrophils, lymphocytes and macrophages[1]. Understanding key factors that underpin disease progression in COVID-19 requires animal models that accurately reflect the pathological characteristics of the disease.

During postmortem examination, macroscopic assessment of inflammatory/congestive changes within the lung based on the percentage of lung surface affected was assessed by a pathologist blinded to the pig treatment status. Assessment was based on a previously established scoring system[24]. While levels varied (14% to 92%), the results suggested considerable lung pathology associated with SARS-CoV-2 infection in the transgenic pigs, particularly from four DPI onwards (Supplemental Table 4).

Microscopic evaluation of nasal turbinate, trachea and lung tissue demonstrated that a subset of pigs displayed key histopathological features similar to those described in fatal COVID-19 (Fig. 6). These included significant neutrophil and macrophage-rich inflammation in the lungs of infected animals from four DPI, including DAD, oedema, focal, fibrin-rich intravascular thrombi and bronchopneumonia[1]. Samples were taken from multiple regions and suggest that inflammation within the lung was focal in nature, as separate samples from the same lungs displayed disparate levels of pathology. Even within the same tissue section, areas of severe lung inflammation were observed adjacent to less affected regions, possibly due to uneven distribution of the viral inoculum (Supplemental Fig. 3).

Due to the challenges associated with working in CL3 conditions with large animals, it was not possible to inflate the lungs prior to sampling of tissues for histology, limiting the ability to accurately quantify the observed histopathology. However, binary qualitative scoring for the presence or absence of pathology indicates substantial inflammation in the lungs of animals from four DPI, with many of the hallmarks commonly associated with severe COVID-19 (Supplemental Table 5). Variability between animals was also observed. For example, in cohort 2, extensive pathology was observed in P46 and P61, but less so in P60, which expresses lower levels of hACE2.

To characterize the lung immunopathology in infected transgenic pigs, lung sections were stained with immune markers Iba-1, PAX5, and CD3. The results showed expanded alveolar, interstitial and peri-bronchiolar macrophage populations by four DPI, while peribronchiolar and perivascular B and T cell populations were formed by seven DPI (Supplemental Fig. 4). These findings are consistent with the pattern of immune cell infiltration observed in fatal COVID-19[1].

### Infection leads to seroconversion in hACE2 pigs

Previous studies have used pigs to model immune responses to COVID-19 vaccines[25]. However, subsequent challenge studies are not possible in WT pigs. The hACE2 pigs represent a highly attractive model for vaccine studies as efficacy in vaccine protection can be determined following challenge. Characterising the immune response to SARS-CoV-2 infection in the transgenic pigs will provide valuable base line data for future vaccine studies. As this was a proof of principle study, the infected pigs were maintained up to seven DPI, limiting a full characterisation of the adaptive immune response which usually occurs seven to ten DPI. However, while full characterisation of the adaptive immune response was outside the scope of the current study, total immunoglobulin (Ig) levels were determined by enzyme-linked immunosorbent assay (ELISA). Although considered early for antibody production, a strong antibody response was detected in animal P53 which was culled seven DPI (Fig. 7). More moderate responses were also detected in P43 at seven DPI and P60 at four DPI, although neither reached the score considered positive (S/P ratio >60%, see methods for details). None of the WT pigs seroconverted. Given the viral load, it is likely that all transgenic pigs would have seroconverted if the course of infection had run beyond the seven days measured. Longer challenge studies will be required to fully characterise the immune response to SARS-CoV-2 infection in the transgenic pigs. However, the data presented suggests the model will be useful for characterising vaccine efficacy.

### hACE2 expression levels may correlate with disease outcome

The use of lentiviral delivery for the generation of transgenic pigs resulted in variable hACE2 expression levels. In vitro studies on primary fibroblast cells indicate that hACE2 expression levels correlate with susceptibility to SARS-CoV-2 replication. If the same concept applies following in vivo challenge studies, levels of hACE2 determined by ear biopsies may be used to predict severity of disease following infection with SARS-CoV-2. Using normalised ranking scores, hACE2 expression in each tissue was compared to virus RNA levels, histology and clinical outcomes. Figure 8 shows a heatmap, comparing the scores, ranked according to hACE2 IHC quantification. The analysis suggests a trend towards increased viral loads and more severe clinical and histological outcomes in pigs with higher levels of hACE2 expression. However, there are several caveats to this analysis. The relatively low numbers of animals used in the challenge study mean statistical significance cannot be achieved and variation in individual animals, unrelated to hACE2 expression, will confound potential correlation. Furthermore, scores will be biased depending on when the animals were culled. For example, peak viral loads occurred at four DPI. Higher levels of inflammation are also more likely to occur at later time points following accumulation of host responses to virus infection. Larger studies will be required to determine whether there is a statistical correlation between hACE2 expression levels and disease severity.

## Discussion

The lack of large animal models that faithfully reproduce the pathology of severe COVID-19 has impeded progress in understanding the underlying mechanisms that drive the inflammatory processes causing disease. A previous study reported the generation of hACE2 transgenic pigs by inserting the hACE2 cDNA downstream of the porcine ACE2 promoter[26]. Although cells generated from those pigs displayed increased susceptibility to SARS-CoV-2 infection, no challenge studies were reported.

Here, we describe the generation of a transgenic porcine model of COVID-19 that is susceptible to infection with SARS-CoV-2 and demonstrates histopathology consistent with moderate to severe disease. Unlike current animal models, infected hACE2 pigs displayed the full range of common clinical signs of COVID-19, including fever, coughing, sneezing, respiratory distress and key pathological signatures in the lung, making these hACE2 pigs a unique model for COVID-19.

Further studies are required to determine if the hACE2 pigs could be utilised to model severe COVID-19. Interpretation of severity based on clinical signs is subjective[4–6]. In humans, loss of pulmonary function causes systemic hypoxemia, characterised by a blood oxygen saturation level of less than 90%, culminating in multi-organ failure, and ultimately, a fatal outcome when emergency interventions fail. Direct measurements of respiration and blood oxygen saturation would help to provide quantitative data indicating the severity of disease in the pigs. The main aim of this study was to determine susceptibility of the transgenic pigs and measurements of blood oxygenation levels were not feasible without specialised equipment, rapid blood tests or extended periods of restraint of animals in CL3 conditions that could not be justified for a proof of principle study. However, the potential use of implantable microchips that would allow real-time measurement of a range of physiological responses, including temperature, blood oxygenation and respiratory rate will be considered for future experiments[27]. Longer time courses would also provide a more accurate determination of disease severity as such symptoms often occur in humans during the second week of infection. An extended time course would also enable a comprehensive characterisation of the cellular and humoral responses to SARS-CoV-2 infection in the pigs, enabling comparative analysis with human responses.

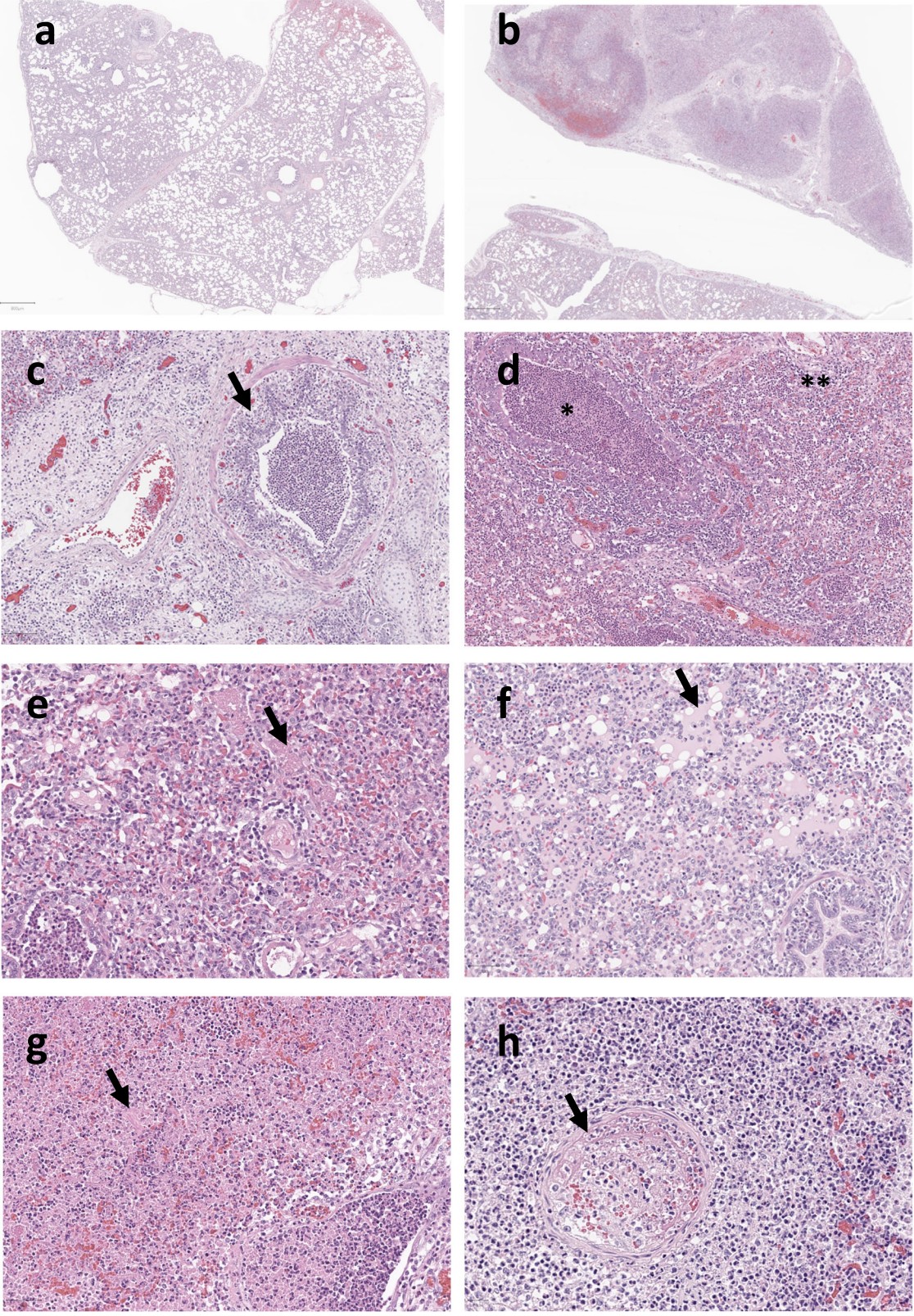

**Fig. 6 | SARS-CoV-2 induces pulmonary inflammation in hACE2 transgenic pigs similar to human COVID-19. a** WT control infected and (**b**) hACE2 infected pigs. In hACE2 pigs there was evidence of neutrophil-rich bronchial inflammation (**c**), widespread bronchiolar (*) and alveolar (**) inflammation (**d**) with associated intra-alveolar fibrin (**e**), oedema (**f**) and parenchymal necrosis (**g**). Occasional organising (arrow) and fibrin-rich thrombi were present in medium calibre vessels (**h**). Representative images shown. **a** WT2 – WT control animal; (**b**–**h**) P46 – hACE2 infected animal, both culled 4 DPI with SARS-CoV-2. (**a**, **b** – 1x magnification; **c** – 10x; **d**–**h** – 20x).

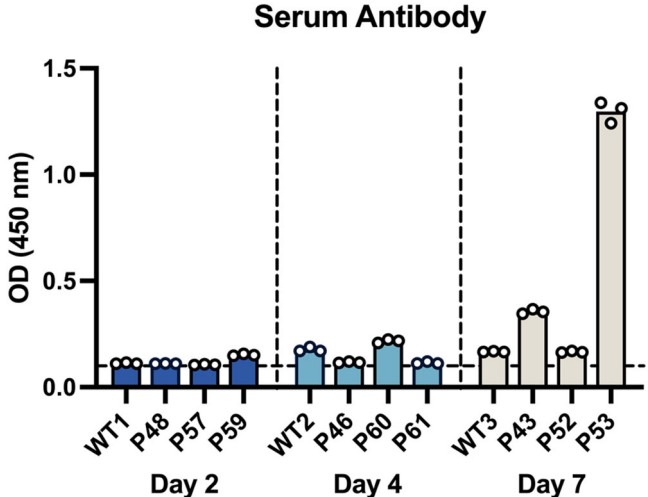

**Fig. 7 | Antibody response to SARS-CoV-2 infection in hACE2 pigs.** Serum was generated from blood samples taken immediately postmortem. Antibody levels were determined using a double antigen multi-species Elisa assay, recognizing SARS-CoV-2 N antigen. Despite the relatively early time point for seroconversion, high levels of antibody were detected in the serum of P53, with lower levels detected in P43 and P60. Data represents the mean from three technical repeats. Source data are provided as a Source Data file.

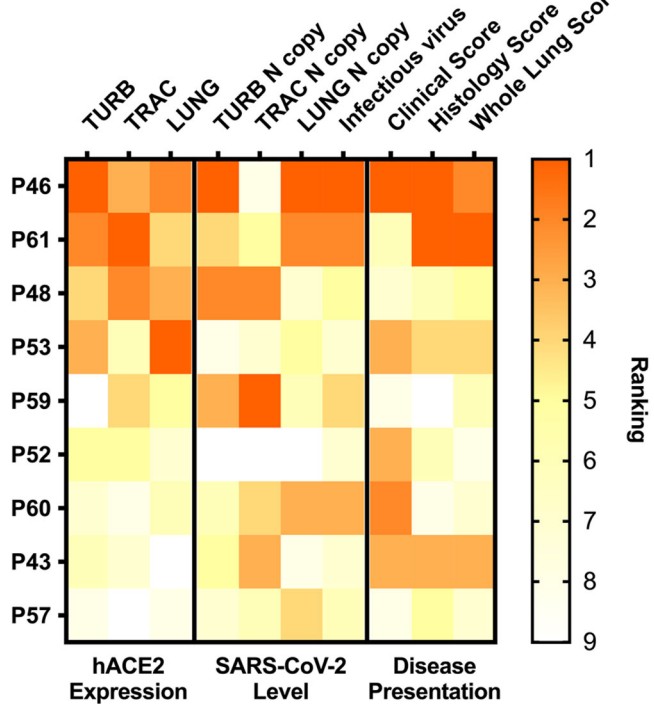

**Fig. 8 | Heatmap displaying correlation between hACE2 expression, SARS-CoV-2 RNA level and disease presentation.** Data were normalised by a rank-based system. Pigs ranked higher in a data category refer to a stronger phenotype or higher value in that specific category and vice versa. Note that pigs can be ranked equivalently in some data categories. The heatmap is ordered by the average pix-elwise H-score for hACE2 expression in the nasal turbinate (TURB), trachea (TRAC) and Lung (LUNG). Source data are provided as a Source Data file.

Fatal outcomes resulting from SARS-CoV-2 would indicate a high level of severity. However, for animal welfare, regulatory and safety reasons, the study was designed to specifically avoid this outcome. Pigs displaying moderate clinical signs, such as respiratory distress, were included in the next time point for culling, reducing the potential for fatal outcomes. It would also be necessary to confirm that death was due to COVID-19 pathology. In some rodent models, aberrant expression of hACE2 in the brain leads to viral encephalitis following infection with SARS-CoV-2[28].

In vitro studies showed higher levels of hACE2 expression correlated with increased SARS-CoV-2 replication and comparative analysis of in vivo data suggested a trend of correlation between hACE2 expression in the tissues of pigs and disease severity. For example, histopathology revealed many hallmarks of severe COVID-19 in the lungs of pigs that expressed the highest levels of hACE2. These results suggest that pigs expressing high levels of hACE2, which can be determined using a simple ear biopsy, could represent accurate models of severe disease. Furthermore, substantial inflammation, despite low levels of virus, indicates that pathology in the lungs of infected pigs is driven by a dysfunctional host response, rather than damage caused by virus replication. This is also a key hallmark of severe COVID-19 in patients[1] and further reflects the similarity in the anatomy, physiology and immune responses of humans and pigs. However, additional studies will be required to confirm that higher levels of hACE2 expression correlate with increased disease severity in the pig model.

High levels of infectious virus in the upper airways, along with observed coughing and sneezing, suggests that airborne transmission between transgenic pigs would likely occur. Rapid diagnosis with existing LFT tests and onset of clearly observable clinical signs provide a potentially powerful model of airborne transmission. Such studies will be critical for evaluation of vaccines and their ability to block transmission, a goal that is yet to be achieved.

Co-circulation of new variants of SARS-CoV-2 and seasonal Influenza virus A (IAV), as well as the threat of avian IAV, has raised significant concerns on the potential impacts of co-infection on vulnerable individuals and the population as a whole[29]. As pigs are naturally susceptible to IAV, this new model will be hugely valuable

for investigating potential consequences of co-infection on disease progression, clinical outcomes, airborne transmission and vaccine and antiviral efficacy. Finally, cross-breeding hACE2 pigs with established porcine biomedical models of underlying co-morbidities, such as obesity and diabetes[30] would leverage additional impact, while cross-breeding with Cas9 pigs[31] would generate a powerful COVID-19 disease model for in vivo and ex vivo precision gene editing.

## Methods

### Plasmids

pLenti CMV GFP Hygro (656-4) was a gift from Eric Campeau & Paul Kaufman (Addgene plasmid #17446; https://www.addgene.org/17446/)[32]. The coding region of human ACE2 (hACE2) was directly synthesized (Life Technologies) and cloned downstream of the CMV promoter to generate pLenti-CMV-hACE2-hygro.

pSL10-K18-hACE2 was used to generate the transgenic pigs. pSL10 is a modification of pLenti6/V5 D-TOPO (Invitrogen), with the ClaI-KpnI fragment removed and replaced with a cPPT/SV40 immediate early promoter/GFP/OPRE cassette. K18-hACE2 was PCR amplified from pK18-hACE2 (a gift from Paul McCray (Addgene plasmid #149449; https://www.addgene.org/149449/)[33]). The amplified cassette was subcloned into BamHI and SalI sites in pSL10 to create pSL10-K18-hACE2.

### Cell lines and viruses

All cells were maintained in Dulbecco's modified Eagle's medium (DMEM; Sigma-Aldrich #D5796) supplemented with 10% fetal bovine serum (FBS) and 100 μg/ml Penicillin-Streptomycin (Gibco) at 37 °C in 5% $CO_2$. Newborn Swine Kidney cells (NSK; RRID: CVCL_8378)

expressing human ACE2 were generated through transduction with pLenti-CMV-hACE2-hygro.

Primary cells were generated from ear notches of ~1 cm³ from each transgenic pig. Samples were washed in PBS and sliced into <1 mm² sized pieces using scalpel blades. The tissue was incubated in digestion medium (DMEM with 20% FBS, 1X antibiotic/anti-mycotic (Capricorn Scientific), 50 μg/mL gentamicin (Gibco) and 0.5 g/ml lyophilised collagenase I (Merck Life Sciences)) in a T25 flask for 24 h at 37 °C with 5% $CO_2$. The sample was mechanically disrupted by pipetting and filtered using a 70 μm cell strainer to produce a single-cell suspension which was cultured with out-growth medium (digestion medium minus collagenase) at 37 °C, 5% $CO_2$ until cells reached confluency.

Details pertaining to the isolation of SARS-CoV-2 viruses were described previously[34]. SARS-CoV-2 variants utilised in this study are EDB2 (early isolate, B.1), EDB-δ-1 (Delta, B.1.617.2) and EDB-o-BA.1–10 (Omicron, B.1.1.529). EDB2 was passaged once in NSK cells stably expressing hACE2 (P2) prior to the challenge study.

### Lentivirus production
All lentiviruses were generated according to the standard protocol for 2nd generation lentiviruses. Each lentiviral plasmid (15 μg) was co-transfected with 12 μg psPAX2 (Addgene #12260) and 3 μg pMD2.G (Addgene #12259) packaging plasmids into Lenti-X 293T cells (Takara) at 70% confluency in T75 flasks using lipofectamine 2000 (Invitrogen #11668019). Supernatant was harvested three days post-transfection and passed through a 0.45 μm filter cartridge, with aliquots frozen at −80 °C. Cells were transduced using one in two dilutions of lentiviral supernatant with fresh media supplemented with 16 μg/ml DEAE dextran.

### Generation of transgenic pigs
Transgenic pigs were generated and genotyped as described previously[35]. Briefly, zygotes were collected by flushing the oviducts of artificially-inseminated super-ovulated Large-White gilts. 50 pl lenti-virus suspension was injected into the perivitelline space, following which 30 zygotes were transferred to the oviduct of each of three identically treated but unmated recipients.

### SARS-CoV-2 infection studies
For infection studies, nine hACE2 transgenic and three wild-type 8-week-old pigs were transferred to CL3 housing. Following a 7-day acclimatisation period, a blood sample was taken by jugular vene-puncture and SARS-CoV-2 (2 ml of a $5 \times 10^5$ $TCID_{50}$/ml suspension; isolate EDB2) was administered intranasally to each pig using a mucosal atomisation device (MAD300). Pigs were monitored twice daily and clinical scores and temperatures recorded. At 2, 4 and 7 DPI, groups of 4 pigs (3 hACE2 transgenic and 1 wild type) were euthanized by overdose of sodium pentobarbital. A blood sample was taken by cardiac puncture and 2 nasal swabs (applied to both nostrils) were taken *postmortem*. The first swab was tested immediately by SARS-CoV-2 lateral flow test (FlowFlex COVID-19 Rapid Antigen Nasal Lateral Flow Test Kit, LO31-118M5), and the second stored in Trizol (Invitrogen) for later analysis. Further tissues were taken at necropsy and stored immediately in 10% buffered formalin, placed in Trizol, or stored in serum-free DMEM. All infection studies were performed at CL3 (BSL3) containment and staff wore appropriate personal protective equipment including FFP3 powered respirator helmets (Pureflo). All animal studies were performed with local and national ethical approval in accordance with the Animal (Scientific Procedures) Act 1986.

### Clinical assessment
The clinical assessment scheme used in this study was established prior to beginning animal work and was approved by our institutional and UK governmental regulatory bodies. Briefly, pigs were monitored for at least 15 min twice daily, by animal care technicians with experience of working with pigs at high containment. They scored each animal against several criteria including demeanour (e.g. response to stimulation, response to presence of food, willingness/ability to stand when provoked), appetite (interest/enthusiasm in feeding/drinking), temperature (rectal temperatures measured twice daily), respiration (degree of effort in respiration, presence/absence of nasal discharge), and other respiratory signs (presence and duration and continuity of coughing and/or sneezing). Within each category, each animal was scored 0, 1, 2 or 3 according to the severity of the feature. At each time point the scores in each category were then added together and the cumulative score for each animal at that timepoint recorded. A cumulative score of 1–6 was regarded as mild clinical severity, 7–10 was regarded as moderate severity and 11–15 was considered severe. This scheme readily allowed animal care staff to identify animals that were at risk of approaching or breaching the regulatory severity limits (humane endpoints) of our Home Office project licence and to seek appropriate veterinary advice.

### PCR
Gene expression was quantified by a 2-step method. RNA was extracted using Trizol following the manufacturer's protocol (Invitrogen) with residual genomic DNA removed by TURBO DNA-free™ Kit (Invitrogen). 1–2 μg RNA was reverse transcribed using the High-Capacity cDNA Reverse Transcription Kit (Applied Biosystems). qPCR reactions were performed using TaqMan™ Fast Advanced Master Mix (Applied Biosystems) and analysed with the Rotor-Gene Q real-time cycler (Qiagen). All gene expression values were normalized to the geometric mean of GAPDH, ACTB and RPL4 and are quantified by the $2^{-\Delta\Delta Ct}$ method unless otherwise specified. Assay ID for TaqMan assay (Applied Biosystems): Hs0108533_m1 (hACE2); Ss03375629_u1 (pGAPDH); Ss03376563_uH (pACTB); Ss03374067_g1 (pRPL4). SARS-CoV-2 N copy number were determined using the SARS-CoV-2 N1+N2 Assay kit (Qiagen, #222015) and standard curve generated from a positive control (Integrated DNA Technologies, #10006625).

Genomic DNA was prepared from ear biopsies of liveborn piglets. Proviral PCRs were performed using the Lenti-X™ Provirus Quantitation Kit (Takara – Cat# 631239) according to manufacturer's protocols.

### Macro- and microscopic assessment of lung injury and inflammation
An estimated percentage of the lung with macroscopically visible pneumonia was recorded for each pig based on a previously described scoring system[24]. Each lung lobe was assigned a number to reflect the approximate volume or percentage of the entire lung represented by that lobe. Ten possible points each were assigned to the right cranial lobe, right middle lobe, cranial part of the left cranial lobe, and caudal part of the left cranial lobe. The accessory lobe was assigned 5 points. The right and left caudal lobes were each assigned 27.5 points to reach a total of 100 points. The total for all the lobes was an estimate of the percentage of the entire visible pneumonia[24].

A defined set of tissue sections was collected from each pig including lungs, trachea sections, turbinates, lymph nodes, tonsil and thymus. Following collection, tissues were immediately immersed into 10% buffered formalin and left overnight before embedding in paraffin blocks. Sections (4 μm) were cut, dewaxed and stained using hematoxylin and eosin. All tissues were qualitatively scored for presence or absence of pertinent histological features. In the lungs, alveolar injury, suppurative pneumonia, alveolar epithelial cell necrosis, bronchiolar necrosis and bronchiolar inflammation was assessed. Alveolar injury was defined as the presence of intra-alveolar oedema, fibrin and/or hyaline membrane formation as well as organising pneumonia and/or type II pneumocyte hyperplasia. A note was made when pleuritis was present.

## Immunohistochemical staining

Immunohistochemical staining was performed on formalin-fixed, paraffin-embedded tissue sections from lung, trachea, turbinates, thymus and tonsil. Where applicable, antigen retrieval was performed in a pressure cooker in a Histos 5 Microwave. Iba1, CD3 and Pax5 were all retrieved using low pH6 Sodium citrate buffer at 110 °C for 5 min, 800 watts (12 min overall). hACE2 was retrieved in a high pH9 buffer solution by Vector Laboratories (H-3301-250) for 50 min at 97 °C full power (51 min overall). SARS-CoV-2 Nucleocapsid protein was retrieved using the high pH buffer solution at 97 °C for 10 min full power. These slides were then allowed to cool for 10 min in cold running water. A rabbit monoclonal antibody (ab108209; ABCAM; 1 µg/ml) for the hACE2 and a mouse monoclonal (B46F; Invitrogen; 1 µg/ml) for SARS-CoV-2 Nucleocapsid protein was used. For immune cell staining, rabbit anti IBa1 (019-19741; Wako; 1 µg/ml), mouse monoclonal PAX5 (AB_398182; Becton and Dickinson; 5 µg/ml) and mouse monoclonal CD3 (NCL-L-CD3-565; Novacastra; 0.2 µg/ml) were used for macrophage, B and T cell staining respectively. Specific Envision secondary antibodies were employed (Envision Rabbit or Envision Mouse) and detection reagents were used for visualisation (Dako Liquid Dab + Substrate Chromogen System). hACE2 positive (P46) and negative sections (P35) based on presence or absence of transgene mRNA expression were used to validate the assay. Pig lymph node was used for PAX5 and CD3 staining controls. Pig brain was used as control material for Iba1, and pig kidney/heart was used as initial ACE2 controls. Negative controls included sections whereby the primary antibody was omitted. Positive control sections for detection of SARS-CoV-2 were from P46, based on RT-qPCR data and P38 uninfected control transgenic pig.

## QuPath implementation of pixelwise H-score

QuPath (version 0.5.1) is an open-source software for digital pathology and whole slide image analysis[36]. The detailed explanation and calculation for the pixelwise H-score is described elsewhere[37]. Briefly, the pixelwise H-score is analogous to the traditional cell-based H-score but is applied to pixels as opposed to individual cells. Using the built in pixel classifier in QuPath, a single user-defined threshold (1 for the IHC images shown here) was set to detect all haematoxylin positive pixels and three separate thresholds, 0.2, 0.15 and 0.1 were set to detect and classify the DAB positive pixels (hACE2) as high, medium and low, respectively.

## Immunofluorescent staining

The porcine primary fibroblast cells were seeded 1-day prior to inoculation with different SARS-CoV-2 variants at an MOI of 3. At 48 HPI, cells were fixed by 10% neutral-buffered formalin (NBF) and permeabilized with 0.1% Triton X100. A conjugated anti-SARS-CoV-2 antibody (AB283243, ABCAM) was used at 0.25 µg/ml for detection of nucleocapsid protein.

## TCID$_{50}$

hACE2-NSKs were seeded in 96-well flat bottom plates 2–3 days in advance to ensure 100% confluency. Each sample was divided into two, which were separately subjected to tenfold dilutions ($10^{-1}$ to $10^{-6}$) in serum-free DMEM. For each sample and dilution, eight replicate wells were used for infection. Culturing medium was removed by aspiration and 50 µl of the diluted virus was added into each well, followed by a 2-h incubation period at 37 °C on a rocker. 100 µl of fresh complete DMEM was added to each well after virus absorption. Cells were incubated for four days and the plates were fixed in 4% paraformaldehyde in PBS for 20 min before being removed from containment facilities for visual inspection of virus-induced cytopathic effect under an inverted microscope. Virus titres were then calculated using the Spearman–Kärber method.

## ELISA assay

All sera were subjected to detection of antibodies against SARS-CoV-2 using ID Screen® SARS-CoV-2 Double Antigen Multi-species (Innovative Diagnostics) according to the manufacturer's protocol. The test detects antibodies to the nucleocapsid protein of SARS-CoV-2 for multiple species (i.e. minks, ferrets, cats, dogs, cattle, sheep, goats, horses and all other receptive species) with a specificity range of 97.8% to 100% as reported by the manufacturer. The assay was validated when the optical density of positive control ($OD_{PC}$) was ≥ 0.35 and at least three times higher than the negative control ($OD_{NC}$). The optical density of each sample ($OD_N$) was used to calculate the S/P ratio value (expressed as %) where $S/P = 100*(OD_N - OD_{NC})/(OD_{PC} - OD_{NC})$. Samples were considered positive if the S/P ratio was greater than 60%, doubtful when ranged between 50 and 60%, and negative when lower than 50%. Optical density (OD) was measured at 450 nm using the Cytation 3 microplate reader (BioTek).

## Reporting summary

Further information on research design is available in the Nature Portfolio Reporting Summary linked to this article.

## Data availability

The authors declare that the data supporting the findings of this study are available within the paper and its supplementary information files. Source data are provided with this paper.

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

## Acknowledgements

We thank Eric Campeau, Paul Kaufman and Paul McCray for sharing reagents. We thank Moredun Bioservices, Robert Bernard and the Large Animal Research and Imaging Facility (University of Edinburgh, United Kingdom) for excellent animal care and members of the Moredun Virus Surveillance unit for technical assistance. This study was funded by UKRI Biotechnology and Biological Sciences (BB/V018922/1), and institutional grants (BBS/E/RL/230002A) as well as a Wellcome ISSF3 award (IS3-R2.26 19/20). The Moredun Research Institute High Containment animal facilities are supported in part by Underpinning National Capacity funding from the Scottish Government Rural and Environment Science and Analytical Services Division (RESAS). We thank Mark Stevens for critical review of the manuscript.

## Author contributions

Conceptualization; F.G., D.J.G., S.L. Methodology; S.L., T.O., S.M., C.W., E.T., C.L.M., G.M.L., S.A., S.B., L.G., T.T., D.M.C. Investigation; F.G., L.F.C., R.B., L.T., C.H.L., T.O. Formal analysis; L.F.C., T.O., J.K.B., D.D., D.J.G., F.G. Resources; C.T.B., S.F. Writing - Original Draft; L.F.C., S.L., T.O., D.D., J.K.B., D.J.G., F.G. Writing - Review & Editing; L.F.C., S.L., T.O., S.M., D.D., J.K.B., D.J.G., F.G. Supervision; S.L., F.G., T.O., G.M.L., D.J.G., F.G. Funding acquisition; T.O., J.K.B., D.J.G., F.G.

## Competing interests

The authors declare no competing interests.

### Ethical approval

Generation and maintenance of transgenic pigs was approved by the Animal Welfare Ethical Review Body of the Roslin Institute, and the infection study was approved by the Animal Welfare Ethical Review Body of the Moredun Research Institute. All experiments involving animals were authorized by the UK Home Office and were performed in accordance with the Animal (Scientific Procedures) Act 1986.
