## [Peer Review file · Nature Communications]

Human ACE2 transgenic pigs are susceptible to SARS-CoV-2 and develop COVID-19-like disease.

Corresponding Author: Dr Finn Grey

Version 0:

Reviewer comments:

Reviewer #1

(Remarks to the Author)

The authors present data related to the development of a human ACE2 transgenic pig model. While humanized pig models have been previously created, authors are the first to report results from a SARS-CoV-2 challenge study.

The methods section is clear and sufficient detail is provided for replication. The clinical outcomes described seem to support the claim that these pigs were infected & experienced mild-moderate disease when challenged with SARS-CoV-2, although only temperature data is provided. As it is presented now, data on viral load and mRNA expression is difficult to assess (see specific comments below).

This model would be an important finding for future coronavirus studies and we recommend this submission be considered for acceptance to publish pending clarification of data (specifically viral load data presented in Fig 1D).

Abstract:

-Per CDC data, the current numbers of positive test results, hospitalizations, mortality events, etc, due to COVID-19 are lower than they've been since the initial identification of the virus. Recommend revising the first few sentences of the abstract to recognize the impact of the pandemic and focus on why animal models for coronaviruses are still needed.

Main:

Line 84 – 32 piglets were born following implantation of oocytes: were there any issues with stillborns, abortions, neonatal mortality? Was there anything different about the 2 gilts in which pregnancy was not successfully achieved?

Line 86 – “Twenty-eight piglets were confirmed as transgenic (data not shown).” Supplemental table 1 shows only 3/32 F0 piglets had provirus not detected (aka: 29 pigs had a Ct value for provirus). Please correct this discrepancy.

Line 92 – F1 cohort in the text is n=30 but supplemental table 2 only has 29 piglets.

-Supplemental table 2: What was the justification for normalizing this data to both GAPDH and P33? Why was piglet P33 chosen as the standard to normalize against? Recommend including this information in the Methods section.

Figures:

Figure 1B:

-Recommend including Cohort 3 patients in this table rather than in the supplemental section.

-Recommend defining Cohorts 1, 2 & 3 in the text and/or in Figure 1. Supplemental table 2 clearly identifies these cohorts but depending on how thoroughly, or in what order, the reader digests this information, the label “Cohort 2” in figure 1B is not well defined.

Figure 1D - Recommend revision of this figure:

-Box/whisker plots are more appropriate for this data than bar graphs.

-Showing absolute values of S1/mL or normalizing to a log scale for WT and hACE infected pigs would be more informative than normalized data. One of the main themes of this paper is that this transgenic model can be infected with SARS-CoV-2. In humans viral load in nasal turbinates is reported as low as 10^3 and as high as 10^{11} N1 copies/mL early in infection; the data here is important to assess the claim that these pigs are highly susceptible to SARS-CoV-2 infection.

-Similar recommendation as for Fig 1B above: recommend including cohort number in title of graphs in 1D, i.e. "Day 2 (Cohort 1)" or similar, for quick identification of study groups by the reader.

Figure 2D: From this magnification this could just as easily be separated blood; recommend increasing magnification to convince the reader this is truly a thrombus.

Supplemental figure 1A: the scale on this graphic is difficult to interpret: hACE2 expression would be expected to be lower than a GAPDH housekeeping gene but one would then expect the scale to be in fractions rather than negative numbers? Recommend changing how this data is normalized/presented so it is easier to understand.

Methods:

-Tracheal sections – please specify location sampled (proximal, mid, distal, etc)

General comments:

-Were these analyses run on tonsils?

-Recommend adding a short discussion on anatomical location and expression levels of ACE2 in humans and comparing to the expression location/levels in this transgenic model.

Reviewer #2

(Remarks to the Author)

Strengths

1. This will be an important animal model to understand both short term and long-term impact of COVID.
2. The work is well carried out and the data presented is rigorous and supports that the ACE2 transgenic pigs can be infected with the early pandemic isolate SARS-CoV-2 (EDB2).
3. There is convincing evidence of viral infective and pathological changes in the lung.

Weaknesses

1. Minor weakness. Line differences. The method used, lentiviruses in zygotes, while highly efficient is also stochastic in nature and leads to line differences something not discussed in this paper. Don't quite understand why they did not use gene editing and place the ACE2 in the K18 locus, or replaced the pig ACE2 with human ACE2 as has been done in mice.

Major weaknesses

1. Lung pathology should be quantitated. How much of the lung was affected. Degree of impact? Airway? alveolar damage?
2. Clinical measurement beside temperature, coughing need to be added. SpO₂, PaO₂/FiO₂, respiratory rate, thrombotic disease, etc. Without that data is difficult to determine whether this is indeed a good model.
3. Immune response data is lacking. Nk cell, T-cell, monocyte, etc data is not presented so it's impossible to compare to normal human responses. Nor whether it would be a good model for vaccine development and testing.

Summary.

While there is nothing wrong with this manuscript it also lacks key data to properly evaluate it as a good model for human COVID.

Reviewer #3

(Remarks to the Author)

The study authors performed a tremendous amount of work to develop a human ACE2 transgenic pig model for the study of severe SARS-CoV-2 infection in humans. These findings represent a significant step to further understanding COVID-19 through pig models; however, clarification on clinical disease and histopathology are necessary to support that moderate to severe clinical disease was accomplished.

Key results: The authors developed a human ACE2 transgenic pig model that was described to develop moderate to severe COVID-19 disease and used multiple methodologies (cell culture, lateral flow tests, PCR, clinical disease, histopathology) to confirm susceptibility to SARS-CoV-2. This provides a new tool in the field's ability to study SARS-CoV-2 infection in an animal model that is more relevant to human disease.

Validity: The methodology of the scoring of clinical disease and histopathology was not detailed enough to understand the authors' interpretation of findings and allow other researchers using this model to reproduce results. Please see below for detailed comments.

Originality and significance: This work represents novel research and is important to more accurately improve our understanding of SARS-CoV-2 infection in humans compared to other models. This study is significant, because there is a need for a genetically tractable animal model that recapitulates human disease for the study of COVID-19 disease, with available experimental tools and that is not cost prohibitive.

Data and methodology: Materials and methods were appropriate to demonstrate successful development of a transgenic model; however, further elucidation in multiple areas is necessary. Please see line edits for more detailed questions and suggestions.

Appropriate use of statistics and treatment of uncertainties: The statistical tests provided were sufficient to demonstrate results.

Conclusions: Additional detail in the conclusions to interpret data (e.g., why cells from certain pigs were not susceptible to infection, certain pigs developed more severe disease than others, etc.) and address study limitations (e.g., utilized one strain of SARS-CoV-2, short study period, etc.) would improve this manuscript.

Suggested improvements: To strengthen this study, the relative amount of hACE2 expression can be compared to other reference genes to demonstrate sufficient quantity in pigs. Moreover, challenging cell cultures with multiple strains of SARS-CoV-2 will verify that this model is appropriate for new variants. Analysis of relative hACE2 expression compared to clinical disease and histopathologic findings would be helpful to explain your results.

Please review the line edits for more detailed instructions for amending this manuscript to address concerns outlined in the assessment.

Main Text

Line 59: Please describe and include a reference for the case definitions of mild, moderate, and severe human COVID-19, and specify what lesions and clinical signs would support that definition. Pulmonary inflammation is a very broad term, and inflammation of different components of the lung have different disease implications.

Line 83: Please explain why the Keratin 18 promoter was used in transgenic pig generation. Is Keratin 18 adequately expressed in pig fibroblasts, as it is primarily expressed in single-layer epithelium?

Line 86: Please provide supplemental figures similar to Supplemental Table 2 regarding how the piglets were confirmed to be transgenic.

Line 102: Please explain why this isolate was chosen for the inoculum. Did it result in more severe disease in humans or was it more prevalent in your geographic area?

Line 106: Have these lateral flow tests been validated for the pig? Please provide references.

Figure 1: As temperatures were not as consistently elevated in cohorts 1 and 3, it would be more transparent to include the average temperature of each cohort in one graph for Figure 1, then include all of the individual temperatures in Supplemental Figure 2. Please include a horizontal line to demonstrate the cut-off for determining if a pig is considered febrile for the reader's ease of understanding.

Line 117: Please specify that the viral RNA was detected in the lung on days 4 and 7.

Line 122: Images provided in Figure 1F do not demonstrate "extensive expression of hACE2". Please replace with more representative photomicrograph or modify interpretation.

Line 123: How was the level of hACE2 RNA interpreted to determine there were "high" levels present? Please indicate in Supplemental Figure 4.

Line 124: Photomicrographs provided in Figure 1F do not demonstrate significant detection of SARS-CoV-2 in the trachea. Please replace with more representative photomicrograph or modify interpretation.

Line 128: Figures 2C and D demonstrate bronchiolar inflammation, not bronchial. "Bronchial" is used throughout the paper to describe inflammation and necrosis in the lung. If Figure 2 represents what was interpreted as inflammation and necrosis in the lung, it should be corrected when mentioned, or a different photomicrograph demonstrating bronchial inflammation and necrosis should be included.

Line 129: It is difficult to determine if the material demarcated with an arrow in Figure 2D is a true fibrin thrombus. Please clarify in statement the frequency at which fibrin thrombi were observed, as it is an important criterion of disease with human COVID-19 infection. In Figure 2D, please include a higher magnification inset of the fibrin thrombus or include a photomicrograph with a more substantial fibrin thrombus.

Figure 2D: See above comments about bronchiolar vs bronchial and the fibrin thrombus.

Line 161: Please include any limitations or interpretations of study limitations, including why the cells from P52 were not susceptible to SARS-CoV-2 infection (as mentioned in Supplemental Figure 1), or if this model will be appropriate for multiple strains of SARS-CoV-2. It would also be helpful to include interpretations of why certain pigs developed more severe disease (is there a correlation between clinical and histopathological disease with relative hACE2 expression?), and if there is a concern that the *Staphylococcus hyicus* outbreak may have contributed to disease in any of the pigs. We appreciate your transparency in enclosing the outbreak, but it is important to explain why it was considered incidental to your

experimental results.

Line 226: The inoculum dose is different from what was included earlier (line 101). Please clarify what was the true experimental dose.

Line 230: Were the two nasal swabs collected one per nostril or were both nostrils sampled with each swab? Depending on methods, the amount of viral RNA may vary between samples.

Line 244: Please include the thermocycler conditions and/or a reference to each protocol. Additionally, please verify the efficiency of GAPDH Δ ACE versus the standard protocol.

Line 245: Were hACE2 gene expression values normalized to other reference genes besides GAPDH? If so, please include in the "PCR section" and in supplemental data. Including multiple reference genes is recommended to meet minimum information for quantitative real-time PCR experiments (MIQE) guidelines to ensure quality data. Please refer to the following publication for additional guidance:

Grätz, Christian, et al. "Obtaining reliable RT-qPCR results in molecular diagnostics—MIQE goals and pitfalls for transcriptional biomarker discovery." *Life* 12.3 (2022): 386. Doi: 10.3390/life12030386

Line 252: Please indicate if positive and negative controls were included in each PCR run. Were PCR products sequenced to ensure there was no contamination between samples and positive controls?

Line 259: Were the tissues scores averaged between the pathologists if there was any disparity, or was there concordance in all of the tissue scores? Additionally, severity is very subjective between pathologists. Please include what qualifies as mild, moderate, and severe (e.g., 0-20% of the lung affected was considered mild, 20-60% was considered moderate, etc.) as supplemental materials.

Line 261: Please replace "protein presence" with "proteinaceous edema" if you are referring to edema within the lung. Additionally, there were other histologic findings evaluated and included in Supplemental Table 3 that should be included in this statement to be consistent throughout, such as hyaline membrane formation, necrosis, pleuritis, and suppurative pneumonia.

Line 262: See comment on line 128; Figure 2C and D demonstrates bronchiolar inflammation not bronchial. Please correct if appropriate.

Line 263: Please specify what cells are necrotic (bronchiolar vs. bronchial epithelial cells, nasal turbinate mucosa vs. glandular epithelial cells, tracheal mucosa vs. glandular epithelial cells) and what type of inflammation (lymphocytic, lymphoplasmacytic, suppurative, etc.). Additionally, inflammation and lymphoid depletion in the lymphoid tissues is not addressed elsewhere in the manuscript, either in the main body of the article or in supplemental materials. If there were lesions in the lymphoid tissues or other significant lesions in non-respiratory tract organs, please include as supplemental materials, or please state in the text that there were no significant lesions.

Line 269: Please outline complete procedure (such as method of antigen retrieval) and reagents (including antibody concentrations and secondary antibody) or include reference to protocol. Replication of immunohistochemistry technique is crucial to interpreting the results of this and future studies.

Line 270: Please expand what tissues and animals were the source of the positive and negative sections.

Supplemental Material

In general: Throughout the supplemental materials, you refer to the wild type pigs as non-transgenic and vice versa. Please use one term for these pigs throughout for consistency or include both.

Supplemental Figure 2A: Please include a horizontal line to demonstrate the cut-off for determining if a pig is considered febrile for the reader's ease of understanding.

Supplemental Figure 2B: If possible, also adding a column that included relative hACE2 expression would help give potential context to why certain pigs developed more severe disease.

Line 395: The clinical assessment scheme is not outlined in the main body of the text or in the supplemental materials. It is crucial to understand what clinical signs were demonstrated at what time point and the severity of those clinical signs to support that this is a good model for severe COVID-19. Is the score based on severity, frequency, amount of time the clinical sign lasted, the number of times the clinical sign occurred? How was demeanor determined and graded? Potentially including a behavioral ethogram may also help readers understand findings and make this study more repeatable.

Line 400: What are the clinical assessment score ranges for mild, moderate, and severe disease? Please also include a key to the colors for Supplemental Figure 2B.

Line 446. Please add "and wild type pigs". Alternatively, refer to previous "in general" comment.

Line 471: Please include “(wild type)” after non-transgenic pigs. Alternatively, refer to previous “in general” comment.

Supplemental Table 1: Thank you for including both tables A and B. If the pig reference number is not helpful information that contributed to its selection process, you can eliminate table A.

Line 502: The “Lenti-X Provirus Quantitation Kit” was not included in methods. Please include in methods and include the manufacturer.

Supplemental Table 2: Please include a key to the colors for hACE2 expression.

Supplemental Table 3: Alternatively, refer to previous “in general” comment about wild type pig nomenclature. It would be helpful to include a row that averaged the score for each experimental cohort (not including the wild type). If possible, also adding a column that included relative hACE2 expression would help give potential context to why certain pigs developed more severe disease.

Line 529: Please replace “Histology” with “Histopathology”.

Line 535: Severity is very subjective across pathologists. Please include what qualifies as mild, moderate, and severe (e.g., 0-20% of the lung affected was considered mild, 20-60% was considered moderate, etc.). Defining the scoring system will allow for consistent interpretation and increases reproducibility of work.

Line 536: Please include the distribution for each type of lesion. This is crucial for understanding the scale of the lesion to further interpret severity of disease.

Reviewer #4

(Remarks to the Author)

Reviewer #5

(Remarks to the Author)

Version 1:

Reviewer comments:

Reviewer #1

(Remarks to the Author)

The edits and additional data provided strengthen the message of this paper and we recommend the manuscript be accepted for publication. A few minor suggestions are included below:

- In general: box/whisker plots provide more information than bar graphs
- Line 272-273 clarification: apical epithelial cells lining bronchioles?
o Vessels are lined by a single layer of endothelial cells
- Line 296: spread through airborne transmission
- Fig 7: Please specify – this is IgG?

Reviewer #2

(Remarks to the Author)

All my concerns have been properly addressed. While the authors were not able to carry out some of the required data collection, they provide logical explanations as to the reasoning and modified the manuscript to emphasize that this is an initial characterization of the model at this time.

However, they have successfully demonstrated that the animals generated are infected by COVID and develop pathological abnormalities and some clinical symptoms, supporting that they can be a good model.

Reviewer #3

(Remarks to the Author)

The authors' efforts to expand methods to support their interpretations improved understanding and repeatability of this

study. The authors made considerable strides to be transparent about their findings, interpretations, and limitations. Suggestions for minor improvements (listed below) will be sufficient for publication.

Main Text

Line 69: Please replace "x-rays" with radiographs.

Line 194: Please replace "wild-type" with WT, as previously defined.

Line 231: "Mild respiratory" signs are given here; however, they are not in Supplemental Table 3. Which is correct?

Line 255: Replace "mucus producing cells in globular glands" with "serous gland epithelium". These glands produce more serous secretions whereas goblet cells produce mucus.

Line 256: "Serous glands" would be more appropriate than "apocrine gland", and it would be more consistent throughout the respiratory tract sections you examined.

Line 257: For consistency throughout, consider referring to glands as "serous gland".

Line 271: Replace "mucus producing cells in globular glands" with "serous gland epithelium".

Line 272-3: Replace "epithelial alveolar cells" with "type I pneumocytes". It is important to distinguish between the two types of cells that can line the alveoli, because they can indicate regeneration, stage of infection, etc. Also, replace "apical epithelial cells" with endothelial cells. To be clear, you are referring to the cells lining blood vessels not the cells lining the bronchioles? These are two different cell populations and have different implications in disease.

Line 286: If you are making this comparison, P59 shows higher staining in the lung than in the nasal turbinates and trachea. Not sure this is significant, though, since we can't see the raw data.

Line 289-91: This theory does not explain clinical severity in P60 but low hACE2 staining.

Line 296: Assuming you meant "spread" here?

Lines 328-9: Please be more specific about where the staining was (ie. which type of cells - bronchial, bronchiolar, type 1 or 2 pneumocytes). Also, where in the cell is the staining (ie. membranous, cytoplasmic, nuclear).

Line 346: The stain evaluated is hematoxylin and eosin, not just hematoxylin.

Figure 6: Thank you for expanding your histopathology sections! It truly supports your interpretation of reflecting true COVID-19-like disease in this model and addresses many concerns in the previous draft.

Line 381: Specify if the macrophages were within alveoli (intra-alveolar) or within alveolar septa. Alveolar macrophages and intravascular macrophages have been shown to have distinct roles in other diseases, specifically in pigs.

Figure 8: Excellent figure to summarize and related all your findings together!

Line 433: Thank you for addressing this study's limitations. This study was an important and significant undertaking despite some limitations.

Line 618-9: Thank you for outlining the grading scheme for determining pathology scores. The scores make much more sense now and this study will be more easily repeatable in the future. Please clarify, however, whether the scoring was really done on "grossly visible" lesions, which means at necropsy. This is what the text seems to say, in which case the title needs to be changed to "Whole lung lesion assessment and histopathology" or something similar.

Line 632-633: It's unclear what "alveolar injury" and "parenchymal necrosis" represent. For alveolar injury, would that include hyaline membranes, intraluminal edema, alveolar histiocytosis, type II pneumocyte hyperplasia, type I pneumocyte necrosis, etc.? For parenchymal necrosis, are you referring to type I pneumocyte necrosis vs the other epithelial cells that you defined?

Lines 637-50: This section is a bit messy. Please define the wattage of "full power", add degrees Celsius to temperatures, and provide concentrations versus dilutions for the antibodies. Also, on line 650, we are assuming you mean "PAX 5 and CD3 controls"?

Line 659: Please replace dilution with concentration.

Supplemental figure 3: Please replace "Gross inflammatory pathology" with "Histopathology". "Gross" would be photos of the tissues at necropsy.

Supplemental table 3: Please add DPI to the left column for ease of the reader.

Reviewer #4

(Remarks to the Author)

Reviewer #5

(Remarks to the Author)

We thank the reviewers for their time and constructive comments. To accommodate the reviewer recommendations, we have reformatted the manuscript from a short communication to a full article. We have addressed each comment either through additional experiments or clarification in the text of the manuscript. We believe the changes made, as suggested by the reviewers, has strengthened the manuscript.

REVIEWER COMMENTS

Reviewer #1 (Remarks to the Author):

Comment: *Per CDC data, the current numbers of positive test results, hospitalizations, mortality events, etc, due to COVID-19 are lower than they've been since the initial identification of the virus. Recommend revising the first few sentences of the abstract to recognize the impact of the pandemic and focus on why animal models for coronaviruses are still needed.*

Response: We agree and have changed the abstract as suggested.

Comment: *Line 84 – 32 piglets were born following implantation of oocytes: were there any issues with stillborns, abortions, neonatal mortality? Was there anything different about the 2 gilts in which pregnancy was not successfully achieved?*

Response: There were no issues around farrowing that deviated from our expectations associated with natural mating. Following our surgical transfer of embryos to surrogate dams, our pregnancy rate is typically between 50% and 60%. We have no explanation as to why some surrogates become pregnant and some do not, but this phenomenon has been consistent over many years of lentiviral transgenesis at Roslin.

Comment: *Line 86 – “Twenty-eight piglets were confirmed as transgenic (data not shown).” Supplemental table 1 shows only 3/32 F0 piglets had provirus not detected (aka: 29 piglets had a Ct value for provirus). Please correct this discrepancy.*

Response: We apologies for the confusion. P28 scored negative by non-quantitative PCR but positive by proviral qPCR, albeit with a high Ct (29.75). As both assays target the lentiviral DNA, we now deem the initial PCR as redundant and have removed the statement.

Comment: *Line 92 – F1 cohort in the text is n=30 but supplemental table 2 only has 29 piglets.*

Response: Again, thank you for pointing out the inconsistency. P41 died three days after birth. Text has been added to line 139 clarifying this.

Comment: *Supplemental table 2: What was the justification for normalizing this data to both GAPDH and P33? Why was piglet P33 chosen as the standard to normalize against? Recommend including this information in the Methods section.*

Response: P33 was chosen as an arbitrary low expressing base line to generate reasonable numbers for easy comparison of hACE2 levels. Choosing the lowest

expressor resulted in higher expressors having relative expression in the thousands. The rational has now been added to the figure legend.

Comment: *Recommend including Cohort 3 patients in this table rather than in the supplemental section. Recommend defining Cohorts 1, 2 & 3 in the text and/or in Figure 1. Supplemental table 2 clearly identifies these cohorts but depending on how thoroughly, or in what order, the reader digests this information, the label “Cohort 2” in figure 1B is not well defined.*

Response: We agree and have included the results and text to clearly define the cohorts in the main text (line 195 and figure 2).

Comment: *Figure 1D - Recommend revision of this figure:*

-Box/whisker plots are more appropriate for this data than bar graphs.

-Showing absolute values of S1/mL or normalizing to a log scale for WT and hACE infected pigs would be more informative than normalized data. One of the main themes of this paper is that this transgenic model can be infected with SARS-CoV-2. In humans viral load in nasal turbinates is reported as low as 10^3 and as high as 10^{11} N1 copies/mL early in infection; the data here is important to assess the claim that these pigs are highly susceptible to SARS-CoV-2 infection.

-Similar recommendation as for Fig 1B above: recommend including cohort number in title of graphs in 1D, i.e. “Day 2 (Cohort 1)” or similar, for quick identification of study groups by the reader.

Response: We agree and have now changed the graphs to box/whisker plots and repeated the PCRs and based the results on copy number, rather than relative values, using an assay targeting the N gene of SARS-CoV-2. This assay consistently detected viral RNA in the lung samples, in contrast to the previous assay using the spike PCR, suggesting this assay is more sensitive. We have included the cohort labelling for clarity as suggested.

Comment: *Figure 2D: From this magnification this could just as easily be separated blood; recommend increasing magnification to convince the reader this is truly a thrombus.*

Response: We have now included a clearer, higher magnification example of a thrombus, in figure 6h.

Comment: *Supplemental figure 1A: the scale on this graphic is difficult to interpret: hACE2 expression would be expected to be lower than a GAPDH housekeeping gene but one would then expect the scale to be in fractions rather than negative numbers? Recommend changing how this data is normalized/presented so it is easier to understand.*

Response: We have removed the RT-qPCR data on hACE2 levels and instead included quantification of hACE2 staining by immunohistochemistry for reasons explained below (see response to reviewer 3, first comment).

Comment: *Tracheal sections – please specify location sampled (proximal, mid,*

distal, etc)

Response: In each case proximal, mid and distal tracheal sections were taken. However, the sections were pooled for subsequent analysis. Therefore, we have not commented on aspects of sample location in relation to trachea.

Comment: *Were these analyses run on tonsils?*

Response: We have IHC data for additional tissues including tonsil sections. However, as we don't have data such as virus load for these tissues, we have not included the analysis in this manuscript and instead have focused on the respiratory tissues. We plan to do more detailed characterisation of other tissues in the future.

Comment: *Recommend adding a short discussion on anatomical location and expression levels of ACE2 in humans and comparing to the expression location/levels in this transgenic model.*

Response: Thank you for the suggestion. We have now included a short discussion on the expression of ACE2 in human tissue and how it compares to the transgenic pigs in the results section (line 241).

Reviewer #2 (Remarks to the Author):

Comment: *1. Line differences. The method used, lentiviruses in zygotes, while highly efficient is also stochastic in nature and leads to line differences something not discussed in this paper. Don't quite understand why they did not use gene editing and place the ACE2 in the K18 locus, or replaced the pig ACE2 with human ACE2 as has been done in mice.*

Response: We believe the approach of replacing the pig ACE2 with human ACE2 may not be effective as expression levels and cellular distribution likely play an equally important, or even dominant role, over the coding sequence of ACE2. Porcine ACE2 has previously been shown to be an effective receptor for SARS-CoV-2 cell entry. Replacing the endogenous porcine ACE2 with human ACE2 has been used and published (referenced in the manuscript) and while hACE2 transgenic pigs were generated, no *in vivo* challenge data was presented. We suspect the pigs were not susceptible to SARS-CoV-2 due to the reasons explained above.

Insertion of hACE2 in the K18 region could disrupt expression of the porcine K18 gene, causing adverse effects. Furthermore, while it is technically feasible that targeted insertion of the ACE2 gene in the K18 locus could be achieved by direct manipulation of zygotes, the expected efficiency of this approach with current gene editing tools is very low. It was thus ruled out from a 3Rs (reduction, refinement, replacement) perspective. The alternative editing strategy would involve manipulating primary cells *in vitro* then performing Single Cell Nuclear transfer (SCNT) to produce piglets. This approach is beset by ethical issues concerning the efficiencies of founder production, the epigenetic status (and thereby health) of the founder generation, and the breadth of data that can be gleaned from a clonal animal population that is intended to model genetically diverse humans. From a scientific,

welfare and practical perspective we therefore decided that a range of hACE2 expression levels in an outbred animal population would be most likely to generate a model with broad utility. Using our approach we have clearly shown that animals with a range of hACE2 expression profiles all become infected with SARS-CoV-2 and display clinical signs consistent with the human condition. This data would not have been evident if we had used a SCNT approach to produce our cohort.

Comment: *Lung pathology should be quantitated. How much of the lung was affected. Degree of impact? Airway? alveolar damage?*

Response: We have now included additional representative images for lung histology of each animal and selected better representative images of specific lung pathology (Figure 6 and supplemental figure 3). We have also included scoring for gross surface inflammation recorded during postmortem, providing quantitative data on how much of the lung was impacted. However, due to the constraints associated with large animal studies in CL3 conditions, we were unable to inflate the lungs prior to sampling and fixation. This made accurate quantitative scoring of the histology images challenging. Following further discussions with the pathologists (TO and DD) rather than report analysis with low confidence, a qualitative, plus minus scoring approach has now been implemented (supplemental figure 5). Text has been included in the manuscript acknowledging this limitation (line 359). Future studies will be designed to enable inflation of the lungs for more accurate scoring of histology. While acknowledging this, it is clear from the current analysis that many of the hallmarks of severe COVID-19 lung pathology are replicated in the transgenic pigs.

Comment: *Clinical measurement beside temperature, coughing need to be added. SpO₂, PaO₂/FiO₂, respiratory rate, thrombotic disease, etc. Without that data is difficult to determine whether this is indeed a good model.*

Response: We agree that such data would be valuable. However direct measures of SpO₂, PaO₂/FiO₂ would require equipment that was not available in the CL3 facility at the time of the challenge study, and we would not have been able to justify addition of such equipment before confirming that the transgenic pigs were susceptible. Oximeter readers were considered, however extended periods of animal restraint would be required for an accurate reading, or repeated anaesthetising of the animals, which couldn't be justified on animal welfare grounds and increased risk to those restraining the animals. We are currently investigating the use of implantable microchips for real time physiological read outs, including temperature, respiratory rates and blood oxygen levels. Text has been included in the discussion to acknowledge the limitations of the current study (line 473).

Comment: *Immune response data is lacking. Nk cell, T-cell, monocyte, etc data is not presented so it's impossible to compare to normal human responses. Nor whether it would be a good model for vaccine development and testing.*

Response: Again, we agree with the reviewer that this data would contribute to the full characterisation of the model. However, the main goal of this initial study was proof of principle, to determine whether human ACE2 transgenic pigs are susceptible to SARS-CoV-2 infection. As this had not previously been reported we could not be sure of the outcome of the study. Furthermore, as we could only justify a seven-day

time course at this stage, a full characterisation of the cellular immune response would not be possible as full responses would take longer than seven days to occur. Furthermore, detailed characterising of immune populations would need to be attempted soon after blood samples were taken. Postmortems took on average 3-4 hours for each time point, with a further 1-2 hours of tissue processing after exiting the CL3. We therefore decided that characterising immune responses were outside the scope of this study.

However, we have now included immunohistochemical staining for macrophage, B and T cells, as well as ELISA data characterising the antibody response. While the data reflects the early time point for the adaptive immune response, one pig, P53, did seroconvert, suggesting the transgenic pigs are likely to be a valuable model for vaccine studies. We intend to perform longer challenge studies in the future that will enable full characterisation of the immune response and discuss the limitations of this proof of principle study for characterising the adaptive immune response in the discussion section.

Reviewer #3 (Remarks to the Author):

Comment: *To strengthen this study, the relative amount of hACE2 expression can be compared to other reference genes to demonstrate sufficient quantity in pigs.*

Response: Our initial experimental set up involved harvesting tissue samples in the large animal CL3 facility. Following completion of the postmortem procedures, samples were transferred to the CL3 molecular biology lab and tissue homogenates generated. These were then aliquoted for RNA extraction and TCID₅₀ assays. For RNA extraction TRIZOL reagent was added immediately to an aliquot of tissue homogenate before freezing at -80°C for processing at a later date. The extracted RNA was entirely used in the initial characterisation experiments, which included quantification of hACE2. Therefore, further RNA was generated from frozen aliquots of tissue homogenates to repeat the assays with additional cellular controls. However, RNA degradation occurred in these samples, possibly due to the freeze thaw cycle without TRIZOL. The sample quality was sufficient for the analysis of viral genome copy number, presumably as the results are based on a logarithmic scale and variability caused by degradation did not impact the interpretation of the data. However, the variability introduced by degradation did impact the results of the human ACE2 levels, making interpretation of the data challenging. While two-to-three-fold changes would not have a major impact on interpretation of the viral data, human ACE2 RT-qPCR data, which is linear in nature, did not correlate with the IHC data for hACE2 levels in the tissues. We therefore quantified the IHC data and used this to compare hACE2 tissue levels between pigs. These issues stem from challenges associated with tissue collection in CL3 conditions where the use of ice and dry ice are problematic. In the future we plan to test various protocols for RNA extraction from tissues to overcome these challenges.

Comment: *Moreover, challenging cell cultures with multiple strains of SARS-CoV-2 will verify that this model is appropriate for new variants.*

Response: We agree and have now included this data (Supplemental figure 1)

Comment: *Analysis of relative hACE2 expression compared to clinical disease and histopathologic findings would be helpful to explain your results.*

Response: We agree that this is an important aspect of the model. However, it is difficult to address this with confidence at this point. This is due to the relatively low number of animals in this study as well as confounding factors, including individual variability intrinsic to *in vivo* studies, especially those using outbred animals. To try and address this we have used a heatmap to compare hACE2 expression levels with viral load, clinical data, temperature and histology (figure 8). While presenting the analysis we have included text to highlight the limitations of this approach.

Comment: *Line 59: Please describe and include a reference for the case definitions of mild, moderate, and severe human COVID-19, and specify what lesions and clinical signs would support that definition. Pulmonary inflammation is a very broad term, and inflammation of different components of the lung have different disease implications.*

Response: Thank you for the suggestion. We have now included this text with references (line 67)

Comment: *Line 83: Please explain why the Keratin 18 promoter was used in transgenic pig generation. Is Keratin 18 adequately expressed in pig fibroblasts, as it is primarily expressed in single-layer epithelium?*

Response: K18 was primarily used as it had previous been successfully used to generate hACE2 transgenic mice that were susceptible to SARS-CoV-2. We tested hACE2 expression from the lentivirus following transduction of Newborn Porcine Tracheal cells (NPT_r) and Newborn swine kidney (NSK) cells. We do not currently have data on expression in fibroblast cells.

Comment: *Line 86: Please provide supplemental figures similar to Supplemental Table 2 regarding how the piglets were confirmed to be transgenic.*

Response: As this PCR and the proviral PCR were both targeting the lentivirus sequence we deemed the initial PCR to be redundant as we only used the proviral data for selection of the breeding pigs. We have therefore deleted this text.

Comment: *Line 102: Please explain why this isolate was chosen for the inoculum. Did it result in more severe disease in humans or was it more prevalent in your geographic area?*

Response: This has now been included (line 187).

Comment: *Line 106: Have these lateral flow tests been validated for the pig? Please provide references.*

Response: As this is the first demonstration of SARS-CoV-2 susceptible pigs, it was not possible until now to validate the lateral flow tests for pigs. However, the results demonstrate they clearly work. As the tests recognise viral antigen they should work

in any species where their use is practical.

Comment: *Figure 1: As temperatures were not as consistently elevated in cohorts 1 and 3, it would be more transparent to include the average temperature of each cohort in one graph for Figure 1, then include all of the individual temperatures in Supplemental Figure 2. Please include a horizontal line to demonstrate the cut-off for determining if a pig is considered febrile for the reader's ease of understanding.*

Response: We have now included all temperature graphs in the main text and included lines to indicate the cut-off for temperatures considered febrile.

Comment: *Line 117: Please specify that the viral RNA was detected in the lung on days 4 and 7.*

Response: Once we switched to using a primer probe set targeting the N gene to enable copy number determination, we were able to detect viral RNA in all lung samples. The text has been adjusted accordingly.

Comment: *Line 122: Images provided in Figure 1F do not demonstrate "extensive expression of hACE2". Please replace with more representative photomicrograph or modify interpretation.*

Response: Additional figures have now been included in figure 4 and supplemental figure 2.

Comment: *Line 123: How was the level of hACE2 RNA interpreted to determine there were "high" levels present? Please indicate in Supplemental Figure 4.*

Response: For the reasons described above the hACE2 RT-qPCR data has now been removed and levels determined by quantification of IHC images.

Comment: *Line 124: Photomicrographs provided in Figure 1F do not demonstrate significant detection of SARS-CoV-2 in the trachea. Please replace with more representative photomicrograph or modify interpretation.*

Response: We have removed the image and stated that while signal was observed, it wasn't sufficiently strong for confident interpretation. Other regions were identified, but all had a similar staining signature. This may be a limitation of the protocol used for trachea or a result of lower levels of viral load in this tissue.

Comment: *Line 128: Figures 2C and D demonstrate bronchiolar inflammation, not bronchial. "Bronchial" is used throughout the paper to describe inflammation and necrosis in the lung. If Figure 2 represents what was interpreted as inflammation and necrosis in the lung, it should be corrected when mentioned, or a different photomicrograph demonstrating bronchial inflammation and necrosis should be included.*

Response: We agree and have now changed the text and figures accordingly

Comment: *Line 129: It is difficult to determine if the material demarcated with an*

arrow in Figure 2D is a true fibrin thrombus. Please clarify in statement the frequency at which fibrin thrombi were observed, as it is an important criterion of disease with human COVID-19 infection. In Figure 2D, please include a higher magnification inset of the fibrin thrombus or include a photomicrograph with a more substantial fibrin thrombus.

Response: We agree and have replace the image with a better example.

Comment: Figure 2D: See above comments about bronchiolar vs bronchial and the fibrin thrombus.

Response: The text has been changed accordingly

Comment: Line 161: Please include any limitations or interpretations of study limitations, including why the cells from P52 were not susceptible to SARS-CoV-2 infection (as mentioned in Supplemental Figure 1).

Response: We have included text addressing the failure to infect P52 cell line (line 168), however at this point we do not know why the cells are not susceptible. They clearly express hACE2 mRNA and the pig they were generated from was susceptible. It's possible the cells have gained a mutation resulting in resistance to the virus, however additional studies will be required to understand this.

Comment: or if this model will be appropriate for multiple strains of SARS-CoV-2.

Response: We have included data in infection of the cell lines with Delta and Omicron strains (supplemental figure 1) and text addressing whether the model will be appropriate for multiple strains of virus (line 169)

Comment: It would also be helpful to include interpretations of why certain pigs developed more severe disease (is there a correlation between clinical and histopathological disease with relative hACE2 expression?),

Response: A section addressing this has now been added (line 416).

Comment: if there is a concern that the *Staphylococcus hyicus* outbreak may have contributed to disease in any of the pigs. We appreciate your transparency in enclosing the outbreak, but it is important to explain why it was considered incidental to your experimental results.

Response: The outbreak occurred in the F0 cohort and animals that showed signs of infection were isolated then culled. No outbreak was observed in the F1 cohort that was used for the challenge study. Therefore, this should not have had any impact on the challenge study or associated data.

Comment: Line 226: The inoculum dose is different from what was included earlier (line 101). Please clarify what was the true experimental dose.

Response: The total dose was given in the main text (1×10^6 TCID₅₀). In the methods section more detail was given to indicate that 2mls of 5×10^5 TCID₅₀/ml which is equivalent to 1×10^6 TCID₅₀.

Comment: Line 230: Were the two nasal swabs collected one per nostril or were both nostrils sampled with each swab? Depending on methods, the amount of viral RNA may vary between samples.

Response: Swabs were applied to both nostrils. The text has been updated (line 578).

Comment: Line 244: Please include the thermocycler conditions and/or a reference to each protocol. Additionally, please verify the efficiency of GAPDH Δ ACE versus the standard protocol.

Response: The thermocycler conditions were set to follow the manufacturers protocol of the qPCR master mix. The TaqMan assays for both GAPDH and hACE2 were purchased commercially and efficiency has been verified to be close to 100%.

Comment: Line 245: Were hACE2 gene expression values normalized to other reference genes besides GAPDH? If so, please include in the “PCR section” and in supplemental data. Including multiple reference genes is recommended to meet minimum information for quantitative real-time PCR experiments (MIQE) guidelines to ensure quality data. Please refer to the following publication for additional guidance:

Grätz, Christian, et al. “Obtaining reliable RT-qPCR results in molecular diagnostics—MIQE goals and pitfalls for transcriptional biomarker discovery.” *Life* 12.3 (2022): 386. Doi: 10.3390/life12030386

Response: We apologise for the lack of details. The thermocycler conditions were set to follow the manufacturers protocol of the qPCR master mix. The TaqMan assays for both GAPDH and hACE2 were purchased commercially and efficiency has been verified to be close to 100%. However, the hACE2 data has now been removed as explained previously.

Comment: Line 252: Please indicate if positive and negative controls were included in each PCR run. Were PCR products sequenced to ensure there was no contamination between samples and positive controls?

Response: For the detection of SARS-CoV-2, a commercially available nucleocapsid plasmid control was included as positive control and uninfected samples as negative control. The hACE2 data has now been removed. Water is included as no template control for each PCR experiment. The region of N gene targeted by the PCR assay is well conserved. Therefore, we don't think sequencing the PCR products would necessarily differentiate between genuine signal and contamination with the positive control.

Comment: Line 259: Were the tissues scores averaged between the pathologists if there was any disparity, or was there concordance in all of the tissue scores?

Additionally, severity is very subjective between pathologists. Please include what qualifies as mild, moderate, and severe (e.g., 0-20% of the lung affected was considered mild, 20-60% was considered moderate, etc.) as supplemental materials.

Response: As described above, due to the challenges in working in CL3 conditions, the lungs were not inflated prior to samples being taken. Because of this the scoring system was changed to a plus minus scheme. Scores of both pathologists have been shown (supplemental figure 5). As severity is subjective, we have changed the text throughout the manuscript, including the title to give less emphasis on whether the model is severe in nature. Instead, we refer to qualitative data that is consistent with hallmarks of severe COVID-19.

Comment: *Line 261: Please replace “protein presence” with “proteinaceous edema” if you are referring to edema within the lung. Additionally, there were other histologic findings evaluated and included in Supplemental Table 3 that should be included in this statement to be consistent throughout, such as hyaline membrane formation, necrosis, pleuritis, and suppurative pneumonia.*

Response: Text in the methods and main section has now been changed accordingly.

Comment: *Line 262: See comment on line 128; Figure 2C and D demonstrates bronchiolar inflammation not bronchial. Please correct if appropriate.*

Response: We have corrected the text accordingly.

Comment: *Line 263: Please specify what cells are necrotic (bronchiolar vs. bronchial epithelial cells, nasal turbinate mucosa vs. glandular epithelial cells, tracheal mucosa vs. glandular epithelial cells) and what type of inflammation (lymphocytic, lymphoplasmacytic, suppurative, etc.). Additionally, inflammation and lymphoid depletion in the lymphoid tissues is not addressed elsewhere in the manuscript, either in the main body of the article or in supplemental materials. If there were lesions in the lymphoid tissues or other significant lesions in non-respiratory tract organs, please include as supplemental materials, or please state in the text that there were no significant lesions.*

Response: The text has been updated to reflect the new scoring approach. As we have focused on pathology in the lung, reference to lymphoid tissue has been removed.

Comment: *Line 269: Please outline complete procedure (such as method of antigen retrieval) and reagents (including antibody concentrations and secondary antibody) or include reference to protocol. Replication of immunohistochemistry technique is crucial to interpreting the results of this and future studies.*

Response: Additional detail has been added to the methods section.

Comment: *Line 270: Please expand what tissues and animals were the source of the positive and negative sections.*

Response: Pig lymph node was used for PAX 5 and CD3. Pig brain was used as control material for Iba1, and pig kidney/heart was used as initial ACE controls. Negatives were sections whereby the primary antibody was omitted. Positive control sections for detection of SARS-CoV-2 were from P46, based on RT-qPCR data and P38 uninfected control transgenic pig.

Comment: *In general: Throughout the supplemental materials, you refer to the wild type pigs as non-transgenic and vice versa. Please use one term for these pigs throughout for consistency or include both.*

Response: Thank you for pointing this out. We have now adjusted the text to refer to control animals as WT.

Comment: *Supplemental Figure 2A: Please include a horizontal line to demonstrate the cut-off for determining if a pig is considered febrile for the reader's ease of understanding.*

Response: This has been added.

Comment: *Supplemental Figure 2B: If possible, also adding a column that included relative hACE2 expression would help give potential context to why certain pigs developed more severe disease.*

Response: We have now included a heat map comparing hACE2 expression levels to clinical scores, temperature and histology scores.

Comment: *Line 395: The clinical assessment scheme is not outlined in the main body of the text or in the supplemental materials. It is crucial to understand what clinical signs were demonstrated at what time point and the severity of those clinical signs to support that this is a good model for severe COVID-19. Is the score based on severity, frequency, amount of time the clinical sign lasted, the number of times the clinical sign occurred? How was demeanor determined and graded? Potentially including a behavioral ethogram may also help readers understand findings and make this study more repeatable.*

Response: An additional section has been added to the methods explaining the clinical assessment scheme. We have also included supplemental table 3 that includes more clinical detail for each pig.

Comment: *Line 400: What are the clinical assessment score ranges for mild, moderate, and severe disease? Please also include a key to the colors for Supplemental Figure 2B.*

Response: This information has now been included in the methods section under clinical assessment scoring and a key has been added to the figure

Comment: *Line 446. Please add "and wild type pigs". Alternatively, refer to previous "in general" comment.*

Response: This data has now been removed – see comments above.

Comment: *Line 471: Please include “(wild type)” after non-transgenic pigs. Alternatively, refer to previous “in general” comment.*

Response: This figure has been replaced with supplemental figure 2.

Comment: *Supplemental Table 1: Thank you for including both tables A and B. If the pig reference number is not helpful information that contributed to its selection process, you can eliminate table A.*

Response: Thank you for the suggestion. We have now deleted table 1B.

Comment: *Line 502: The “Lenti-X Provirus Quantitation Kit” was not included in methods. Please include in methods and include the manufacturer.*

Response: This was included at the end of the methods section for PCR (line 613).

Comment: *Supplemental Table 2: Please include a key to the colors for hACE2 expression.*

Response: The colours are an arbitrary indicator of hACE2 expression levels based on Microsoft auto formatting to give a quick visual aid for ranking. As such there is no preset scale for this. Text has been added to the figure legend to indicate the colour scheme meaning.

Comment: *Supplemental Table 3: Alternatively, refer to previous “in general” comment about wild type pig nomenclature. It would be helpful to include a row that averaged the score for each experimental cohort (not including the wild type). If possible, also adding a column that included relative hACE2 expression would help give potential context to why certain pigs developed more severe disease.*

Response: Supplemental table 3 has now been replaced with Supplemental table 5 for reasons explained above. The scores have been included in the heat map comparing to hACE2 expression.

Comment: *Line 529: Please replace “Histology” with “Histopathology”.*

Response: This has been updated.

Comment: *Line 535: Severity is very subjective across pathologists. Please include what qualifies as mild, moderate, and severe (e.g., 0-20% of the lung affected was considered mild, 20-60% was considered moderate, etc.). Defining the scoring system will allow for consistent interpretation and increases reproducibility of work.*

Response: As indicated above the scoring system has now been changed to a binary presence or absence and reference to severity has been limited to qualitative comparison to histological changes in tissue from fatal COVID-19 cases.

Comment: *Line 536: Please include the distribution for each type of lesion. This is*

crucial for understanding the scale of the lesion to further interpret severity of disease.

Response: As described above, we were unable to inflate the lungs prior to collection of samples. We have therefore decided the best approach is to use a qualitative scoring approach and give examples of each lesion and low magnification examples of tissues from the lungs of each animal. We have however included data indicating a percentage score of surface inflammation following examination of the lungs during postmortem, which provides an indication of the extent of disease in each animal at the selected time point.

We thank the reviewers for their time and constructive comments. We have addressed each comment by editing the manuscript. We have made additional changes to improve clarity or correct minor issues. We have corrected an error in the figures showing temperatures, which misaligned temperature with timepoint. (Fig 3a, b, c). This does not affect interpretation of the data. We have added a panel to Figure 5c, showing an additional section of lung tissue stained for nucleocapsid which adds further detail. Some minor text corrections were also made.

REVIEWER COMMENTS

Reviewer #1 (Remarks to the Author):

Comment: *In general: box/whisker plots provide more information than bar graphs*

Response: We agree and have changed all graphs to include overlaid data points.

Comment: *Line 272-273 clarification: apical epithelial cells lining bronchioles? Vessels are lined by a single layer of endothelial cells.*

Response: Thank you for pointing this out. This has now been changed accordingly.

Comment: *spread through airborne transmission*

Response: This has now been edited.

Comment: *Please specify – this is IgG?*

Response: The kit does not discriminate between IgG and IgM. Instead, it detects total Ig. The text has been updated to clarify this.

Reviewer #3 (Remarks to the Author):

Comment: *Line 69: Please replace “x-rays” with radiographs.*

Response: The text has been updated accordingly

Comment: Line 194: Please replace "wild-type" with WT, as previously defined.

Response: Agreed and changed to WT.

Comment: Line 231: "Mild respiratory" signs are given here; however, they are not in Supplemental Table 3. Which is correct?

Response: As shown in Supp table 3, mild respiratory signs (increased respiratory effort and /or occasional cough) were observed in P48, P46, P61, and P53 within the first 48 hours. Additional text has been added to avoid confusion.

Comment: Line 255: Replace "mucus producing cells in globular glands" with "serous gland epithelium". These glands produce more serous secretions whereas goblet cells produce mucus.

Comment: Line 256: "Serous glands" would be more appropriate than "apocrine gland", and it would be more consistent throughout the respiratory tract sections you examined.

Comment: Line 257: For consistency throughout, consider referring to glands as "serous gland".

Comment: Line 271: Replace "mucus producing cells in globular glands" with "serous gland epithelium".

Response: Thank you for pointing this out. We agree with all four comments above and have adjusted the text accordingly.

Comment: Line 272-3: Replace "epithelial alveolar cells" with "type I pneumocytes". It is important to distinguish between the two types of cells that can line the alveoli, because they can indicate regeneration, stage of infection, etc. Also, replace "apical epithelial cells" with endothelial cells. To be clear, you are referring to the cells lining blood vessels not the cells lining the bronchioles? These are two different cell populations and have different implications in disease.

Response: We agree, although as we cannot differentiate between type I and type II we've referred more generically to pneumocytes

Comment: Line 286: If you are making this comparison, P59 is shows higher staining in the lung than in the nasal turbinates and trachea. Not sure this is significant, though, since we can't see the raw data.

Response: We agree and have altered the text to reflect the difference in P59.

Comment: Line 289-91: This theory does not explain clinical severity in P60 but low hACE2 staining.

Response: We agree and have added text to indicate factors other than hACE2 expression likely play a role in determining the level of pathogenesis in individual animals.

Comment: Line 296: Assuming you meant "spread" here?

Response: Yes, thank you for spotting this.

Comment: *Lines 328-9: Please be more specific about where the staining was (ie. which type of cells - bronchial, bronchiolar, type 1 or 2 pneumocytes). Also, where in the cell is the staining (ie. membranous, cytoplasmic, nuclear).*

Response: More detail has now been included in the text.

Comment: *Line 346: The stain evaluated is hematoxylin and eosin, not just hematoxylin.*

Response: We agree and have changed the text.

Comment: *Figure 6: Thank you for expanding you histopathology sections! It truly supports your interpretation of reflecting true COVID-19-like disease in this model and addresses many concerns in the previous draft.*

Response: Thank you for this comment.

Comment: *Line 381: Specify if the macrophages were within alveoli (intra-alveolar) or within alveolar septa. Alveolar macrophages and intravascular macrophages have been shown to have distinct roles in other diseases, specifically in pigs.*

Response: We have now specified this in the text.

Comment: *Figure 8: Excellent figure to summarize and related all your findings together!*

Response: Thank you. This figure was devised by first author Long Fung Chau.

Comment: *Line 433: Thank you for addressing this study's limitations. This study was an important and significant undertaking despite some limitations.*

Response: Thank you. We agree and hope the model can contribute to our understanding of COVID-19 pathogenesis.

Comment: *Line 618-9: Thank you for outlining the grading scheme for determining pathology scores. The scores make much more sense now and this study will be more easily repeatable in the future. Please clarify, however, whether the scoring was really done on "grossly visible" lesions, which means at necropsy. This is what the text seems to say, in which case the title needs to be changed to "Whole lung lesion assesement and histopathology" or something similar.*

Response: We agree and have altered the text accordingly.

Comment: *Line 632-633: It's unclear what "alveolar injury" and "parenchymal necrosis" represent. For alveolar injury, would that include hyaline membranes, intraluminal edema, alveolar histiocytosis, type II pneumocyte hyperplasia, type I pneumocyte necrosis, etc.? For parenchymal necrosis, are you referring to type I pneumocyte necrosis vs the other epithelial cells that you defined?*

Response: This has now been clarified.

Comment: *Lines 637-50: This section is a bit messy. Please define the wattage of "full*

power", add degrees Celsius to temperatures, and provide concentrations versus dilutions for the antibodies. Also, on line 650, we are assuming you mean "PAX 5 and CD3 controls"?

Response: Apologies, this section has now been updated.

Comment: *Line 659: Please replace dilution with concentration.*

Response: This has now been done.

Comment: *Supplemental figure 3: Please replace "Gross inflammatory pathology" with "Histopathology". "Gross" would be photos of the tissues at necropsy.*

Response: This has now been corrected.

Comment: *Supplemental table 3: Please add DPI to the left column for ease of the reader.*

Response: We have now added DPI.